# Therapeutic homology-independent targeted integration in retina and liver

Patrizia Tornabene[1,2,9], Rita Ferla [1,2,9], Manel Llado-Santaeularia [1,9], Miriam Centrulo[1], Margherita Dell'Anno [1,2], Federica Esposito [1], Elena Marrocco[1], Emanuela Pone [1,2], Renato Minopoli[1], Carolina Iodice[1], Edoardo Nusco[1], Settimio Rossi[3], Hristiana Lyubenova [1], Anna Manfredi[4,5], Lucio Di Filippo[5], Antonella Iuliano[1], Annalaura Torella[1,6], Giulio Piluso [6], Francesco Musacchia [1], Enrico Maria Surace[2], Davide Cacchiarelli[4,7], Vincenzo Nigro[1,6] & Alberto Auricchio [1,8 ✉]

Challenges to the widespread application of gene therapy with adeno-associated viral (AAV) vectors include dominant conditions due to gain-of-function mutations which require allele-specific knockout, as well as long-term transgene expression from proliferating tissues, which is hampered by AAV DNA episomal status. To overcome these challenges, we used CRISPR/Cas9-mediated homology-independent targeted integration (HITI) in retina and liver as paradigmatic target tissues. We show that AAV-HITI targets photoreceptors of both mouse and pig retina, and this results in significant improvements to retinal morphology and function in mice with autosomal dominant retinitis pigmentosa. In addition, we show that neonatal systemic AAV-HITI delivery achieves stable liver transgene expression and phenotypic improvement in a mouse model of a severe lysosomal storage disease. We also show that HITI applications predominantly result in on-target editing. These results lay the groundwork for the application of AAV-HITI for the treatment of diseases affecting various organs.

[1] Telethon Institute of Genetics and Medicine (TIGEM), 80078 Pozzuoli, Italy. [2] Medical Genetics, Department of Translational Medicine, Federico II University, 80131 Naples, Italy. [3] Eye Clinic, Multidisciplinary Department of Medical, Surgical and Dental Sciences, University of Campania L. Vanvitelli, 80131 Naples, Italy. [4] Telethon Institute of Genetics and Medicine (TIGEM), Armenise/Harvard Laboratory of Integrative Genomics, 80078 Pozzuoli, Italy. [5] Next Generation Diagnostic Srl, 80078 Pozzuoli, Italy. [6] Department of Precision Medicine, University of Campania L. Vanvitelli, 80138 Naples, Italy. [7] Department of Translational Medicine, Federico II University, 80131 Naples, Italy. [8] Medical Genetics, Department of Advanced Biomedical Sciences, Federico II University, 80131 Naples, Italy. [9] These authors contributed equally: Patrizia Tornabene, Rita Ferla, Manel Llado-Santaeularia. ✉email: auricchio@tigem.it

Adeno-associated viral (AAV) vectors are the most broadly used for in vivo gene therapy applications; this is due to their favorable safety profile, wide tropism, and ability to provide long-term transgene expression. The retina and the liver are among the most relevant target tissues of AAV-mediated gene therapy[1]. The first FDA-approved gene therapy for an inherited disease was an AAV vector administered subretinally to patients with a rare form of childhood blindness[2], while systemic administrations of AAV vectors that target the liver are in various phases of clinical development for hemophilias (OMIM 306700, OMIM 306900) and several inborn errors of metabolism like ornithine transcarbamylase deficiency (OMIM 311250), familial hypercholesterolemia (OMIM 143890) and mucopolysaccharidosis type VI (MPS VI, OMIM 253200)[3]. However, AAV-mediated liver and retina gene delivery still present challenges.

In total, 30–40% of cases of retinitis pigmentosa (RP) have an autosomal dominant pattern of inheritance with *Rhodopsin* (*RHO*) being the most commonly mutated gene (RP4, OMIM 613731). *RHO* mutations are responsible for about 25% of autosomal dominant RP cases in the United States and about 20% of cases elsewhere in the world[4]. P23H is the most common *RHO* mutation in North America due to a founder effect of European origin, representing 9% of all cases of autosomal dominant RP in the United States[4]. P23H RHO does not fold properly and accumulates in the endoplasmic reticulum (ER), which results in ER stress and ultimately in photoreceptor cell death[5,6]. Several other *RHO* mutations have gain-of-function (GOF) effects. In these cases, conventional gene supplementation therapy is ineffective.

One of the major limitations of AAV-mediated gene therapy is the episomal nature of recombinant AAV genomes which undergo dilution in proliferating tissues, e.g., during hepatocyte division in liver which is the main target of AAV systemic administration[7,8]. In fact, newborn animals treated with intravenous AAV present a rapid and progressive decline in liver transgene expression[7,9–11], which prevents their application to newborn-toddler patients. In the clinical trial of AAV-mediated liver gene therapy for MPS VI that we are currently conducting (ClinicalTrials.gov number NCT03173521), patients under 4 years of age are excluded for this reason. In addition, liver injury followed by regeneration might cause loss of AAV-mediated transgene expression in adult patients[8,12].

Versatile genome editing with CRISPR/Cas9 has the potential to overcome AAV gene transfer limitations. In the retina, CRISPR/Cas9 gene delivery is being evaluated to obtain allele-specific knockout of common *RHO* GOF mutations[13–16]. However, this approach is limited by the availability of specific guide RNA (gRNA) and protospacer adjacent motif combinations for specific Cas9 targeting of the GOF allele. In addition, this approach is mutation-specific, which limits its clinical applicability due to the high RP4 locus heterogeneity[17–20]. Ideally, any mutant *RHO* should be exchanged with a correct copy without the need for allele-specific approaches. However, homology-directed repair (HDR), which is the most used mechanism for site specific integration[21], is not a viable option because of its relative inefficiency in terminally differentiated photoreceptors. In mouse liver, HDR has been exploited in combination or not with nuclease-induced double-stranded breaks to mediate precise integration of a donor DNA with homology arms to the *albumin* (*Alb*) locus, which is highly transcribed in hepatocytes[22,23], thus overcoming AAV genome dilution in proliferating hepatocytes. This is being tested in the clinic for therapy of hemophilia B (NCT02695160) and mucopolysaccharidoses type I (NCT02702115) and II (NCT03041324). However, the efficiency of this HDR-dependent system is still under investigation and the inclusion of homology arms needed for HDR limits the size of the therapeutic sequence that can be packaged in an AAV vector.

Recently, a strategy for DNA integration at a desired locus has been described that exploits non-homologous end joining (NHEJ), a DNA repair pathway more diffused and active than HDR, which could therefore achieve improved efficiency in both retina and liver. This homology-independent targeted integration (HITI) uses a donor DNA that is flanked by the same gRNA/Cas9 target sequences within the gene of interest. After Cas9 cleaves both the gene and the donor DNA, the NHEJ machinery of the cell integrates the donor DNA during the repair of the DNA double-strand breaks, with surprisingly high rate of precise integration[24,25]. Inverted integration of the donor DNA is avoided by inverting its gRNA target sequences, so that Cas9 can recognize and cut the target sequence again if inverted integration occurs[26]. HITI precision and off-target integration need to be further characterized in order to establish this as a safe alternative to HDR. Moreover, further investigation of its therapeutic potential in terminally differentiated neurons, like photoreceptors, and in dividing tissues, like the liver, is necessary.

Here, we propose an adapted HITI with a donor DNA that includes both STOP codons to block the expression of the endogenous gene and a translation start site followed by a correct copy of the therapeutic transgene (Fig. 1). Specifically, we developed a system based on two AAV vectors. One AAV encodes *Streptococcus pyogenes* Cas9 (SpCas9) under the control of a tissue-specific promoter, to restrict SpCas9 expression to the targeted tissue. The other AAV includes: (1) a gRNA expression cassette, in which the expression of a specific gRNA (or a scramble-gRNA, as a negative control) is driven by the U6 promoter and (2) the donor DNA which contains the coding sequence (CDS) for the therapeutic gene which is preceded by a translation start site (START), followed by the bovine growth hormone polyadenylation sequence and flanked by inverted gRNA target sites identical to the site selected to cleave the endogenous gene (Fig. 1). As NHEJ-mediated integration of the donor DNA will likely generate INDELs, inclusion of STOP codons in all three open reading frames upstream of the START will block expression of the endogenous gene, regardless of the integration precision (Fig. 1).

In the retina, we propose to use HITI to replace both the P23H mutant and wild-type *Rho* alleles, thus providing an allele-independent genome editing strategy for RP4. In the liver, we target the *Alb* locus in newborn, proliferating hepatocytes of a mouse model of the severe lysosomal storage disease MPS VI to achieve stable therapeutic levels of arylsulfatase B (ARSB), the lysosomal enzyme which is deficient in MPS VI.

## Results

**AAV-HITI in the mouse and pig retina**. To knockout both wild-type and mutant mouse *rhodopsin* (m*Rho*) and simultaneously replace them with a correct copy of the human *Rhodopsin* (h*RHO*) CDS, we designed a gRNA targeting the first exon of the m*Rho* gene to induce integration of our donor at the beginning of the endogenous gene and thus disrupting its expression. We demonstrated the efficiency of this m*Rho*-gRNA in vitro using a scramble gRNA that does not align with either the mouse or pig genomic sequences, as negative control (Supplementary Fig. 1a and Supplementary Table 1). To determine HITI efficiency, we initially included the CDS for *Discosoma red fluorescent protein* (DsRed) in our donor DNA to act as a reporter. We designed two donor DNAs with different and broadly used translation start sites (STARTs): either a consensus kozak sequence (therefore referred to as kozak-DsRed), or a small synthetic 50 bp internal ribosomal entry site (IRES) sequence (IRES-DsRed)[27]. Since the donor DNA does not include a promoter, we expected to see expression of the reporter only after correct integration in the

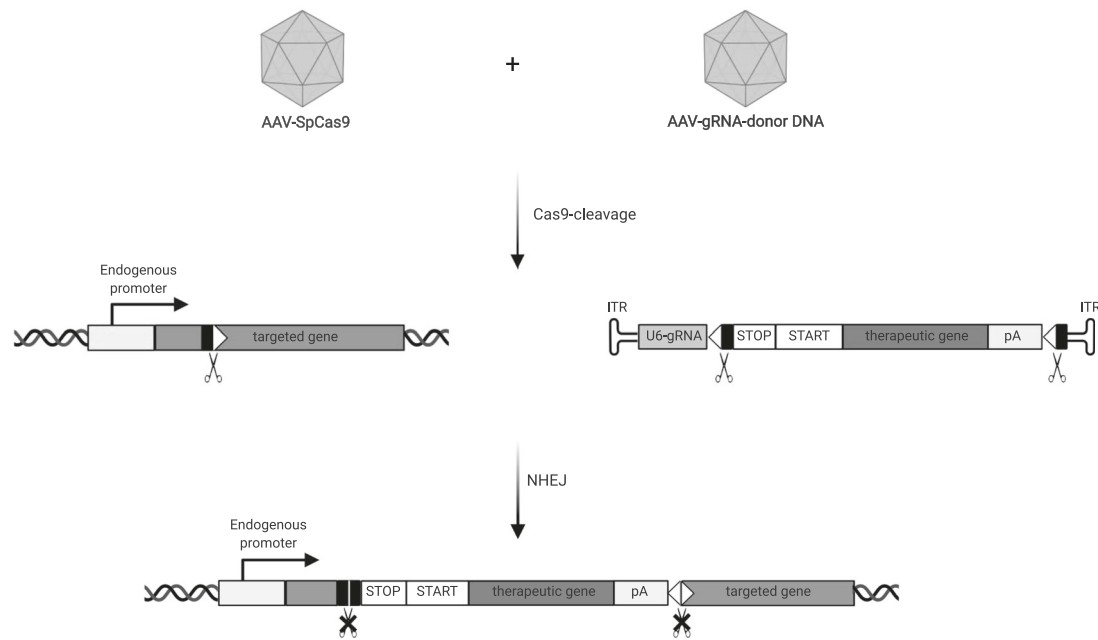

**Fig. 1 Schematic of the AAV-HITI therapeutic approach.** AAV-delivered SpCas9 targets both the endogenous locus of a targeted gene and the inverted gRNA target sites flanking the donor DNA. Non-homologous end joining (NHEJ) recruits the donor DNA at the targeted locus. STOP codons knockout the endogenous gene while START sites initiate translation of the therapeutic gene. AAV adeno-associated viral vector, SpCas9 *Streptococcus pyogenes* Cas9, ITR inverted terminal repeats, U6-gRNA gRNA expression cassette containing the U6 promoter, the gene-specific or scramble gRNA and the gRNA scaffold; black rectangles and white triangles represent the two parts of the SpCas9 target sites flanking the SpCas9-induced double-strand break; scissors represent SpCas9-mediated double-strand breaks; STOP Stop codons in the 3 open reading frames, START translation start site (either kozak or internal ribosomal entry site, IRES), pA bovine growth hormone polyadenylation sequence; crossed scissors represent inability of SpCas9 to recognize target sites after homology-independent targeted integration has occurred.

m*Rho* locus. We tested both the kozak- and the IRES-DsRed constructs in vitro and observed efficient integration and gRNA-specific DsRed expression (Supplementary Fig. 1b, c and Supplementary Results and Discussion). Then, to test this approach in the retina we generated a two AAV vector system: one AAV encoded SpCas9 under the control of the interphotoreceptor retinol-binding protein (IRBP) promoter (IRBP-SpCas9, Supplementary Fig. 2a), while the other AAV carried either the m*Rho*-gRNA (gRNA) or the scramble-gRNA expression cassette together with the donor DNA (either kozak-DsRed or IRES-DsRed, Supplementary Fig. 2b). Four weeks after subretinal injection of AAV vectors in mouse retinas, fluorescence microscopy analysis showed 4.2% of DsRed+ photoreceptors in the injected area when using the gRNA and the kozak-DsRed, and 4.7% when using IRES-DsRed donor DNA (Fig. 2a). No DsRed+ photoreceptors were detected in scramble-treated retinas, confirming the lack of potential leaky expression caused by AAV inverted terminal repeats (ITRs) transcriptional activity. We observed significant variability in the integration efficiency between eyes from the same group, with a single IRES-DsRed+ gRNA-treated eye showing 11.8% of DsRed+ photoreceptors. A thorough analysis of SpCas9 activity at both on- and off-targets sites (Supplementary Table 2) showed that this is mostly accurate, as reported in Supplementary Results and Discussion, Supplementary Fig. 3 and Supplementary Table 3.

Next, we tested whether HITI could be used to edit the genome of photoreceptors in pigs which are a large, widely used pre-clinical model. We designed a gRNA targeting the first exon of pig *Rhodopsin* (p*RHO*), which has a similar efficiency in vitro to the one complementary to m*Rho* (Supplementary Fig. 1a). AAV vectors carrying either the gRNA or scramble expression cassettes with either the kozak-DsRed or IRES-DsRed donor DNAs (Supplementary Fig. 2c) were co-injected subretinally in pigs

with the IRBP-SpCas9 vector (Supplementary Fig. 2a). One month later, we observed 1.2% DsRed+ photoreceptors in the injected area of eyes treated with kozak-DsRed + gRNA and 4.9% in eyes treated with IRES-DsRed + gRNA (Fig. 2b). No DsRed+ photoreceptors were observed in scramble-treated retinas, independently of the donor DNA used. This result shows that HITI is also feasible in a clinically-relevant model of the human eye. We also showed that SpCas9 activity is tissue-specific and efficient (Supplementary Fig. 4a, b and Supplementary Results and Discussion) and detected HITI in about 5% of p*RHO* alleles (Supplementary Fig. 4d and Supplementary Results and Discussion) in line with the histological quantification of DsRed+ photoreceptors.

**AAV-HITI improves the retinal phenotype of a mouse model of autosomal dominant retinitis pigmentosa.** To assess whether AAV-mediated HITI can provide therapeutic benefits in a mouse model of RP4[28], we generated two AAV plasmids, both carrying a donor DNA with IRES and h*RHO*-2A-DsRed CDS with either the gRNA or the scramble expression cassette (Supplementary Fig. 2d). Real-Time PCR analysis showed h*RHO* expression in gRNA-transfected SpCas9-EGFP+/DsRed+ HEK293 cells but not in gRNA- and scramble-transfected SpCas9-EGFP+/DsRed− cells (Supplementary Fig. 5a). Therefore, 7-days old $Rho^{P23H-/+}$ mice received subretinal injections of both the IRBP-SpCas9 vector (Supplementary Fig. 1a) and the h*RHO*-2A-DsRed donor DNA vectors (Supplementary Fig. 1d). Electroretinography performed at approximately post-natal day (p) 120 showed significant improvement of both A- and B-wave amplitudes in the gRNA- compared with the scramble-injected eyes (Fig. 2c, age-matched wild-type C57BL/6J A- and B-wave response at 20 lux = 245.6 ± 10.8 and 517 ± 23.6 μV, respectively). This was mirrored by a significant amelioration of both visual acuity (Fig. 2d, age-matched wild-type

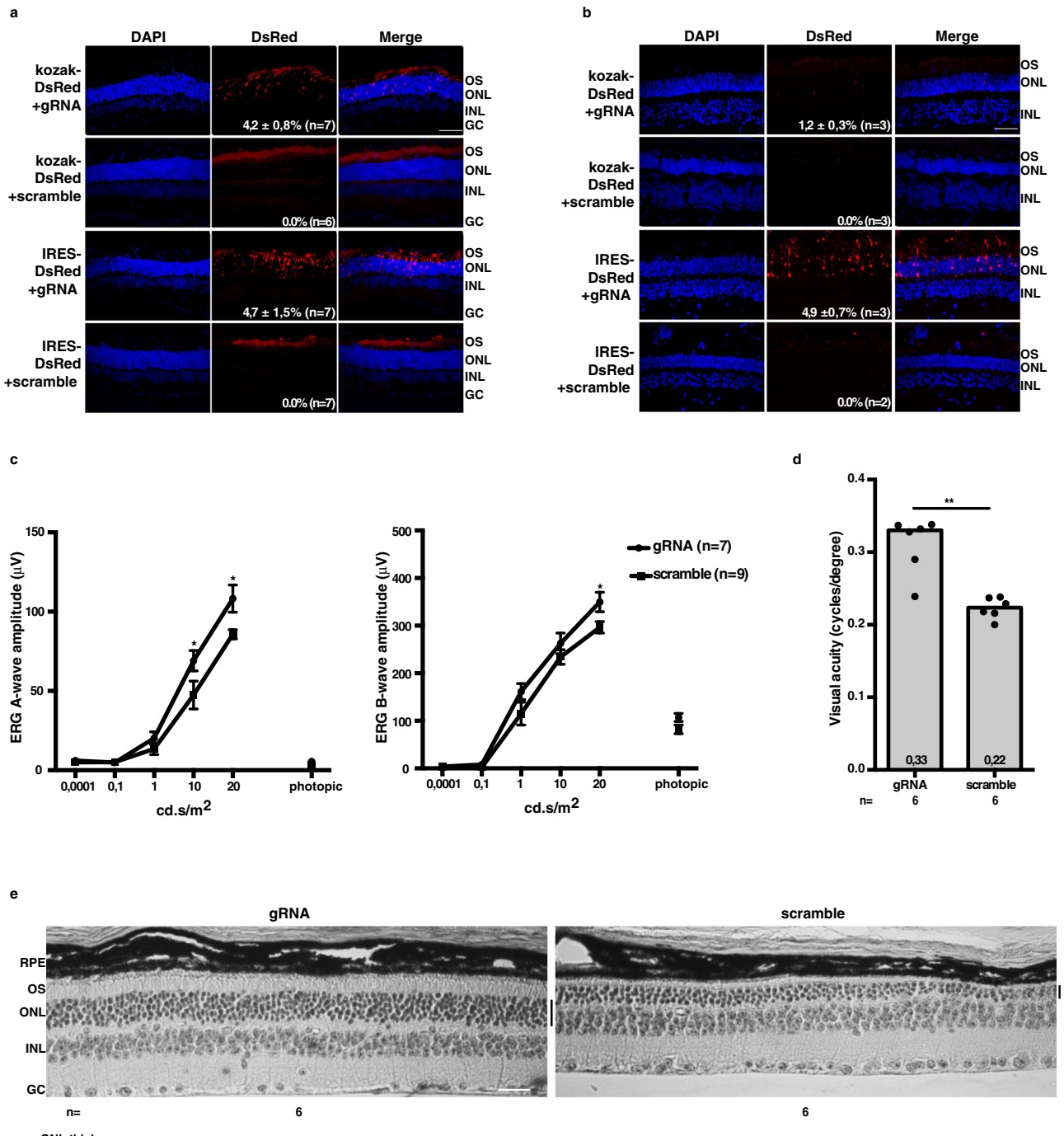

**Fig. 2 Subretinal administration of AAV-HITI results in efficient photoreceptor editing and improves the retinal phenotype of $Rho^{P23H-/+}$ mice. a, b** Fluorescence microscopy representative images of retinal cryosections 30 days after subretinal injection with either $2.5 \times 10^9$ and $2 \times 10^{11}$ genome copies/ eye of each vector in mouse (**a**) and pig (**b**) retina, respectively. Slices were stained with DAPI; the white scale bar in the upper merge image equals 50 μm. **c–e** $Rho^{P23H-/+}$ mice were injected subretinally at post-natal day (p) 7 with $1.7 \times 10^9$ genome copies of each vector and analyzed at around p120 by electroretinography (**c**), optomotry (**d**) and histology (**e**). **c** Electroretinogram A- and B-wave amplitude was measured at constant luminance (cd s/m²) representative of scotopic (0.0001 to 20 cd s/m²) and photopic conditions. **e** The white scale bar in the left image equals 35 μm. Results are represented as mean ± SEM for each group of treatment (numbers in **a**, **b**, line-dot, **c**, **e**) or as a single measurement for each mouse (dot) and as median for each group of treatment (column, **d**). *$p < 0.05$; **$p < 0.01$ (**a**, **b** unpaired $t$-test; **c** either unpaired $t$-test or Wilcoxon rank sum test; **d** Wilcoxon rank sum test; **e** unpaired $t$-test). Further details on statistical analysis, including exact $p$ value, can be found in "Methods" section. Source data are provided as a Source Data file. DsRed *Discosoma red fluorescent protein*, cd s/m² candela-seconds per meter squared, RPE retinal pigmented epithelium, OS outer segments, ONL outer nuclear layer, INL inner nuclear layer, GC ganglion cells, *n* number of biologically independent samples.

C57BL/6J response = 0.4 ± 0.006 cycles/degree) and outer nuclear layer (ONL) thickness (Fig. 2e; age-matched wild-type C57BL/6J ONL thickness = 36.4 ± 0.83 μm), which was evident in the ventral and central sides of the eye which are close to the injection site (Supplementary Fig. 5b).

**AAV-HITI in newborn mouse liver**. To overcome one of the major limitations of AAV-mediated liver gene therapy, i.e., loss of episomal AAV in proliferating tissues, we exploited HITI as a strategy to obtain stable transgene expression in hepatocytes after neonatal AAV delivery. To do this, we designed a gRNA (Supplementary Table 1) that efficiently targets exon 2 of the mouse *albumin* (m*Alb*) gene, which is the nearest targetable sequence to the endogenous promoter (Supplementary Fig. 4a). We then generated a two AAV vector system with one AAV encoding SpCas9 under the control of a liver-specific promoter[29] [hybrid liver promoter (HLP)-SpCas9, Supplementary Fig. 6a] and the other AAV carrying the donor DNA (Supplementary Figs. 6b and 7) containing the reporter DsRed to monitor integration efficiency, as we have done for the retina. Since our donor DNA does not include a promoter, we expected DsRed to only be expressed after integration in the m*Alb* locus.

We compared HITI to conventional gene replacement therapy using an AAV vector which expresses DsRed from the liver-specific thyroxine-binding globulin (TBG) promoter (TBG-DsRed, Supplementary Fig. 6b). Fluorescence microscopy analysis of liver cryosections showed around 2% of DsRed+ hepatocytes in livers treated with gRNA (Fig. 3a), while no DsRed+ hepatocytes were observed in scramble-treated livers. Importantly, the percentage of DsRed+ hepatocytes remained stable from p15 to p90 in the gRNA group, while in livers treated with TBG-DsRed, DsRed+ hepatocytes significantly decreased from 18.2% at p15 to 2.5% at p90 (86% reduction, Fig. 3a). This reduction is consistent with AAV genome dilution during hepatocyte proliferation[7–11].

The characterization of SpCas9 activity as well as of HITI at both the on- and off-target sites (Supplementary Table 4) is reported in Supplementary Results and Discussion, Supplementary Fig. 8 and Supplementary Table 5.

**Systemic administration of AAV-HITI in newborn mice results in stable therapeutic levels of transgene expression from liver**. Next, we tested if HITI at the m*Alb* locus in newborn mice results in stable and therapeutically relevant levels of transgene expression. As a therapeutic transgene we used *ARSB*, the lysosomal hydrolase defective in MPS VI, a rare lysosomal storage disease. MPS VI patients under 4 years of age are excluded from an ongoing phase I/II AAV gene therapy clinical trial we are conducting (ClinicalTrials.gov number NCT03173521), due to the potential loss of transgene expression during hepatocyte division. Since ARSB is secreted in the bloodstream and can be non-invasively measured, it can be used as readout of liver transduction[30]. ARSB deficiency results in abnormal glycosaminoglycan (GAG) storage and urinary secretion, which is a useful biomarker of MPS VI[31]. We generated AAV vectors carrying the donor DNA cassette including the CDS for human ARSB (h*ARSB*), as well as a gRNA expression cassette for either gRNA or scramble as control (Supplementary Fig. 6c). Each of these vectors were systemically co-delivered in combination with the HLP-SpCas9 vector (Supplementary Fig. 6a) in neonatal MPS VI mice (p1-2). Serum ARSB activity was measured in gRNA-treated MPS VI mice at levels that were around 9% of normal (Fig. 3b, normal serum ARSB values are 11,825 ± 334 pg/ml[32]) and remained stable over time (ANOVA $p = 0.632$) up to 1 year after treatment. Serum ARSB activity in scramble-treated MPS VI mice was undetectable ($p < 0.05$ for gRNA vs. scramble at each

timepoint). Normalized ARSB activity was observed at sacrifice in gRNA-treated livers, confirming stability of transgene expression (Fig. 3c).

Importantly, while no significant differences in urinary GAGs were observed between scramble and gRNA-treated groups at p30, AAV-HITI-mediated ARSB expression was able to significantly lower urinary GAGs at p150 and p360 to levels that were not statistically different from those of normal, unaffected mice (Fig. 3d).

GAG storage normalization was observed in liver and myocardium (Fig. 3e). Bone histological analysis showed reduction of vacuolization exclusively in cortical osteocytes of gRNA-treated mice (Fig. 3e). Consistently, biochemical assessment of GAG levels in liver, kidney and spleen showed their normalization in gRNA-treated mice (Fig. 3f-h).

A potential drawback of our current HITI design that targets m*Alb* exon 2 is that it can interfere with albumin secretion. We measured serum albumin levels in gRNA-treated mice at p360 and observed no significant (ANOVA $p = 0.12$) reduction in serum albumin levels compared with scramble-treated mice and untreated controls (Fig. 3i), which is consistent with the low percentage of DsRed+ hepatocytes (Fig. 3a) and the low percentage of INDELs detected (Supplementary Results and Discussion).

## Discussion
Our current work aims to overcome some of the major challenges related to AAV vector-based in vivo gene therapy. In the retina, therapies to treat dominant disease sub-types demand mutant allele knockout, while allele-specific approaches only have limited clinical applicability due to mutation heterogeneity. In the liver, AAV genome dilution during hepatocyte proliferation causes loss of transgene expression, which precludes neonatal treatment that is required for many inborn errors of metabolism. Here we have used a strategy based on HITI and a donor DNA design to achieve targeted integration and disease phenotype correction in both retina and liver.

We have included in the donor DNA two START sites with different characteristics. On one hand, we used the kozak sequence, which is the common signal for ribosomal translation initiation. However, the presence of the kozak sequence from the endogenous gene could compete with the one from the donor DNA. On the other hand, we used a small synthetic sequence, i.e., IRES, which has been shown to efficiently recruit the ribosome[27]. However, since it would be integrated close to the endogenous gene translation start site, we did not know whether it would work efficiently. Our experiments demonstrate that both START signals mediate *Discosoma red fluorescent protein* (DsRed) translation however with some differences. The kozak constructs performed better in vitro than in vivo, this could be due to a number of factors including: (1) the shorter distance between the kozak and the promoter in vitro; (2) the efficiency of ribosome recruitment to the IRES sequence; (3) or photoreceptor-specific differences in protein translation. Additionally, the differences in efficiency of kozak-DsRed and IRES-DsRed donors in pigs compared to mice could depend on the different site of insertion.

In the retina, we showed that HITI is effective in differentiated photoreceptors, the most desirable target for gene therapy of inherited retinal diseases, as the majority of genes mutated in these forms of blindness are indeed expressed in photoreceptors[33]. A previous study had suggested the therapeutic potential of HITI in the retina based on integration occurring in the retinal pigment epithelium[25]. The retinal pigment epithelium, however, is a much less relevant gene therapy target than photoreceptors since the majority of genes mutated in inherited retinal diseases are expressed in photoreceptors[34], which are also more difficult to transduce with

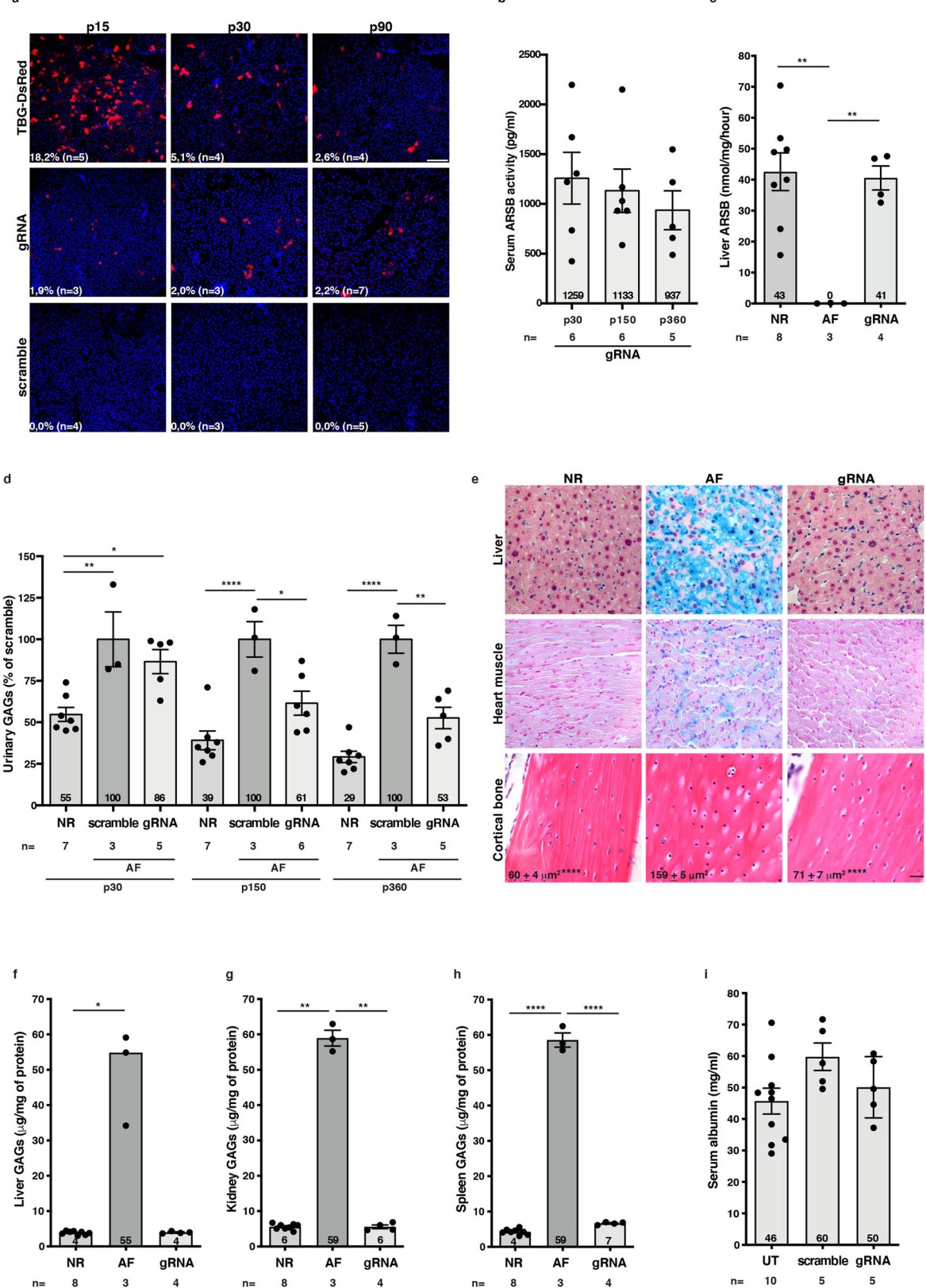

viral vectors than the retinal pigment epithelium. Therefore, our study represents a major advancement over previous observations, as it opens the possibility of targeting a large group of diseases which would benefit from HITI in photoreceptors. Also, this approach is especially relevant in photoreceptors because they are post-mitotic

cells, in which integration via HDR is not an option due to the low activity of this DNA repair pathway[26,35].

Here, we quantified the efficiency of HITI in photoreceptors with two different donor DNA configurations previously tested in vitro. Unlike what we observed in vitro, IRES worked similarly

**Fig. 3 Stable hepatocyte transduction following neonatal administration of AAV-HITI improves the phenotype of a mouse model of MPS VI.**
**a** Fluorescence microscopy representative images of TBG-DsRed-, gRNA- and scramble-treated livers of mice sacrificed at post-natal day (p) 15, p30 and p90. C57BL/6J mice were injected at p1-2 with $4 \times 10^{13}$ genome copies/kg of each vector. Sections were stained with DAPI; the white scale bar in the upper right image equals 100 μm. The percentage of DsRed+ hepatocytes is reported inside each image. **b** Serum arylsulfatase B (ARSB) activity measured in gRNA-treated MPS VI mice is reported. **c** ARSB activity was measured in liver of gRNA-treated and control affected (AF) MPS VI mice as well as of normal (NR) controls. **d** Urinary glycosaminoglycans (GAGs) were measured in gRNA-, scramble-treated MPS VI AF mice and in NR controls. Values are reported as percentage of age-matched scramble controls. **e** Histological representative images of storage in liver, heart muscle and cortical bone of gRNA ($n = 4$)- and control MPS VI AF ($n = 2$, except for cortical bone $n = 4$) mice and of NR ($n = 3$, except for cortical bone $n = 5$) controls; the black scale bar in the lower right image equals 25 μm. GAG levels were measured in liver (**f**), kidney (**g**) and spleen (**h**) of gRNA-treated and control MPS VI AF mice as well as in NR controls. **i** Serum albumin levels were measured at p360 in gRNA-, scramble-treated (NR, $n = 2$; AF, $n = 3$) and untreated (UT) mice (NR, $n = 7$; AF, $n = 3$). MPS VI mice were injected at p1-2 with $6 \times 10^{13}$ genome copies/kg of each vector. Results are represented as median for each group of treatment (numbers in **a** and column, **f**) or as a single measurement for each mouse (dot) and as either mean ± SEM for each group of treatment (numbers in **e** and column, **b**–**d**, **g**–**i**). *$p < 0.05$; **$p < 0.01$; ***$p < 0.001$; ****$p < 0.0001$ (**a**, **f**, Kruskal–Wallis followed by the Conover's all-pairs rank comparison test; **b**–**e**, **h**, **i**, one-way ANOVAs followed by the Tukey post hoc test; **g** Welch ANOVA followed by the Games Howell test). Further details on statistical analysis, including exact $p$ value, can be found in "Methods" section. Source data are provided as a Source Data file. TBG thyroxine-binding globulin, $n$ number of biologically independent samples.

or better than kozak as a translation start site in both mouse and pig retinas, reaching about 5% efficiency in the area of injection.

HITI efficiency in the retina was mostly limited to the area immediately adjacent to the injection site. We hypothesized that this could be due to the low efficiency of photoreceptor co-transduction (about 24%) by two different AAV vectors reported in the mouse retina[36]. As a higher rate of photoreceptor co-transduction (about 73%) has been reported in the pig than in the mouse retina[37], we expected to observe higher HITI efficiency. However, while we show that HITI occurs in pig photoreceptors, which are in a retinal structure closer to the human, its levels were similar to those observed in the mouse retina. Since increasing photoreceptor co-transduction by AAV does not seem to increase HITI, additional post-transduction processes, e.g., the NHEJ repair machinery[26], could be limiting. Indeed, our data suggests that in mouse photoreceptors, double-strand breaks were often corrected through donor DNA integration, while in pig a large proportion were repaired as insertion or deletions (Supplementary Results and Discussion). Importantly, the restriction of HITI to the area immediate to the injection site may not be an issue as preserving the structure and function of a small retinal area is sufficient to preserve vision in several retinal diseases.

Indeed, the levels of HITI obtained following subretinal administration of AAV in a mouse model of autosomal dominant RP drove a significant improvement of retinal function and morphology which however did not reach those of wild-type mice. While this supports the translational potential of this approach for RP4 as well as for other inherited retinal diseases, it also highlights that the efficiency of the method should be further improved.

Further advances in HITI efficiency in the retina may improve observed therapeutic outcomes for RP4 and may also be relevant for gene therapy of other inherited retinal diseases. In this direction, increasing the amount of donor DNA delivered may increase successful integration events. Similarly, using AAV serotypes engineered to have high transduction ability, like those depleted of tyrosine residues in their capsid[38], or other DNA or ribonucleoprotein delivery vectors[33,39–41], could help deliver *Streptococcus pyogenes* (SpCas9) and the donor DNA more efficiently to photoreceptors, and thus increase HITI efficiency in the retina. Alternatively, using other methods for targeted integration, like exploiting the homology-mediated end joining approach proposed by Yao et al.[42], which combines NHEJ and HDR, could increase the efficiency of targeted integration while maintaining the main features of the donor DNA that we have developed. Indeed, Nishiguchi et al. recently showed that microhomology-

mediated end joining is a viable method for targeted integration in photoreceptors[43]. While their method achieved integration in 10% of photoreceptors, it is important to point out that they used a single AAV encoding for a *Staphylococcus aureus* Cas9, the gRNA expression cassette and a small donor DNA (240 bp). This single AAV approach does not depend on co-transduction of photoreceptors by two different AAVs and thus the higher efficiency is to be expected. Our approach, although less efficient in principle, allows integration of a larger donor DNA and, in the best cases, achieved similar levels of overall efficiency to those reported by Nishiguchi et al.[43]. Additionally, new gene editing methodologies like base editing[44–47] and prime editing[48] can be used; both use editing complexes that, like HITI, have to be divided in two AAVs. Additionally, base editing maintains the same limitation as allele-specific gene editing, which is the ability to only target a particular mutation. In the case of prime editing, although integration of external DNA without DNA double-strand breaks is possible, there is a limitation on the size of the inserted fragment, which so far cannot exceed 100 bp[48]. Our approach, in comparison, achieves efficient insertion of a donor DNA up to 2.7 kb long and is therefore able to target all *Rhodopsin* mutations with one single design.

We also demonstrated that neonatal delivery of AAV-HITI achieves stable therapeutic transgene expression in mouse liver with about 2% of DsRed+ hepatocytes. Interestingly, SpCas9 cleavage efficiency in the *albumin* locus was around 10% (Supplementary Results and Discussion). These numbers are very consistent with those reported by Suzuki et al., who used similar vector doses to perform HITI in the liver of neonatal Ai14 mice[25]. This suggests that HITI efficiency may be limited by both liver transduction and NHEJ efficiency. On the other hand, since the ratio between HITI and DNA cleavage efficiency is similar to that reported by others when using HDR[49,50], DNA cleavage also appears an important determinant of integration efficiency in the liver.

We next tested the therapeutic potential of HITI in the liver of a mouse model of MPS VI, a severe lysosomal storage disease, and show that newborn MPS VI mice treated with AAV-HITI have stable serum ARSB levels up to 12 months after treatment, as well as normalized liver ARSB activity. This results in a significant reduction of GAGs in urine and tissues, and of vacuolization in cortical bone osteocytes. This improvement is similar to that reported after treating adult MPS VI mice with either $2 \times 10^{12}$ genome copies/Kg of the canonical AAV-ARSB vector (which is also a dose used in our NCT03173521 clinical trial) or enzyme replacement therapy[30] which is the current standard-of-care for MPS VI.

In addition to overcoming AAV genome dilution due to hepatocyte proliferation, HITI offers other advantages over conventional liver gene therapy with AAV. Several studies indeed showed an increased incidence of hepatocellular carcinomas (HCCs) following liver gene transfer in newborn mice[51–53]. In particular, the work of Chandler et al. clearly showed that the combination of neonatal intravenous delivery of high doses of AAV with strong promoters, like the hepatocyte-specific TBG (used in several clinical trials including NCT03173521) or the constitutive chicken beta actin promoter, cause insertional mutagenesis and HCC in mice[51]. In contrast, the approach we developed uses the weaker HLP for SpCas9 expression (Chandler et al.[51] showed weaker promoters to be non-carcinogenic) and, importantly, no promoter in the donor DNA. Therefore, the risk of HCC development due to HITI-induced insertional mutagenesis should be lower than when using high doses of an AAV vector with a strong promoter. The potential carcinogenesis derived from off-target HITI and AAV integration is discussed in detail in the Supplementary Results and Discussion section. Moreover, AAV genome dilution during hepatocyte division should reduce the copies of the HLP-SpCas9 vector, thus reducing the risk of both insertional mutagenesis and SpCas9 off-target activity. As an alternative, SpCas9 delivery as a ribonucleoprotein using lipid nanoparticles has been shown to achieve very high editing levels in adult rat liver[54] while restricting the time of SpCas9 expression in hepatocytes. That would potentially eliminate the risk of undesired insertion of promoter elements and eventually, further reduce SpCas9 off-target activity. Nanoparticles are also expected to achieve higher transduction in the human liver in comparison to AAVs, and thus should be more relevant for a potential clinical application of this approach.

One of the disadvantages of our HITI approach in the liver might be mild reductions in the levels of circulating albumin due to targeting of its gene CDS. Although circulating albumin was not different in gRNA-treated mice, future targeting of *albumin* 3′ untranslated region in combination with sequences that allow co-expression of the therapeutic transgene with *albumin*[22,23] will eliminate risk of HITI-induced *albumin* knockout. Importantly, HITI in *albumin* could be easily adapted to treat other diseases that require high levels of transgene expression from hepatocytes by simply changing the transgene CDS in the donor DNA without needing to change the gRNA or the donor DNA conformation.

Overall, we have developed a novel approach that exploits HITI and shown its feasibility in both post-mitotic retinal photoreceptors and in dividing hepatocytes in vivo, demonstrating the broad applicability of this technology for genome editing. Even though additional formal safety studies will be required before this approach can be considered for applications in humans, we have provided proof-of-concept evidence of the therapeutic potential of HITI in animal models of dominant conditions or for stable expression of secreted proteins from newborn liver, potentially broadening the patient population that could benefit from in vivo gene therapy.

## Methods

**Study design**. This study was designed to define the efficiency of AAV vector-mediated HITI at correcting genetic diseases in the retina and the liver. Editing efficiency in retina and liver was defined in cryosection fluorescent images by using the ImageJ software to count and calculate the percentage of either photoreceptor or hepatocyte positive to *Discosoma red fluorescent protein* (*DsRed*). DNA obtained from both treated retina and liver were used for detection of insertions and deletions (INDELs) by Tracking of INDELs by Decomposition (TIDE), for next-generation sequencing analysis of on/off-targets of gRNAs, HITI efficiency and precision at the on-target site and, for the liver, off-target HITI, and donor DNA integration by using ad-hoc algorithms. In all in vivo studies, right and left eyes were randomly assigned to each treatment group. In addition, in the studies on the disease models, female and male mice were considered equivalent and randomly

assigned to treatment groups. Littermate controls were used when available. Therapeutic efficacy of HITI in the retina was assessed by evaluating the impact of subretinal delivery of AAV-HITI on the retinal phenotype of an animal model of autosomal dominant RP. Therapeutic efficacy of HITI in the liver was assessed by evaluating the impact of neonatal intravenous delivery of AAV-HITI on the phenotype of an animal model of mucopolysaccharidosis type VI (MPS VI). In both cases, observers were blind to both genotype and treatment of the animals. Sample sizes were determined based on previous experience and technical feasibility.

**Generation of the AAV vector plasmids**. The plasmids used for AAV vectors production derived from a pAAV2.1 plasmid that contains the inverted terminal repeats of AAV serotype 2[55].

The mouse *rhodopsin* (m*Rho*)); pig *Rhodopsin* (p*RHO*) and mouse *albumin* (m*Alb*) gRNAs (Supplementary Table 1) were designed using the Benchling gRNA design tool (www.benchling.com), selecting the gRNAs with the best predicted on-target and off-target scores, targeting the first or second exon of *rhodopsin* and *albumin* genes, respectively. The scramble gRNA was designed to not align with any sequences in the mouse or pig genome. gRNAs were then generated as forward (Fwd) or reverse (Rev) oligonucleotides (Supplementary Table 1), annealed and cloned in the PX458 [pSpCas9 (BB)-2A-GFP] (Addgene #48138) plasmid encoding *Streptococcus pyogenes* Cas9 (SpCas9), as described by the Zhang lab[56]. These plasmids were used for in vitro experiments reported in Supplementary Fig. 1A. Then, the gRNA expression cassettes (including the U6 promoter) were PCR-amplified and subcloned in PCR-Blunt II-TOPO (Invitrogen, Carlsbad, California, United States) before cloning in the pAAV2.1 backbone using In-Fusion (Takara, Kusatsu, Japan).

All donor DNA fragments were generated by PCR amplification adding six STOP codons in all three open reading frames and either the kozak or the IRES[27] sequences, as well as the appropriate 5′ and 3′ gRNA target sites flanking the donor DNA, as PCR primer overhangs. PCR fragments were subcloned in PCR-Blunt II-TOPO (Invitrogen) before cloning in a pAAV2.1 backbone using In-Fusion (Takara).

The *Discosoma red fluorescent protein* (DsRed) donor DNAs were generated by PCR amplification of the DsRed CDS and the bovine growth hormone polyadenylation (pA) sequence from a DsRed plasmid published in[57]. PCR fragments were subcloned in PCR-Blunt II-TOPO (Invitrogen) before cloning in a pAAV2.1 backbone using In-Fusion (Takara).

The pAAV2.1 plasmid, bearing m*Rho* P23H, under the control of the cytomegalovirus (CMV) promoter, and the pA sequence (CMV-m*Rho* P23H) and used in Supplementary Fig. 1c, was described in a previous publication[58]. The plasmid p946 (IRBP-SpCas9, Supplementary Fig. 2a) bearing SpCas9 under control of the IRBP and a synthetic poly A (spA) was generated by cloning the IRBP promoter in the commercial PX551 plasmid (pAAV-pMecp2-SpCas9-spA, Addgene #60957) using HindIII and AgeI restriction sites by conventional ligation.

To generate the donor plasmids for in vitro experiments (Supplementary Fig. 1c), the DsRed donor DNAs for m*Rho* were cloned into the pAAV2.1 plasmid encoding the reporter *enhanced green fluorescent protein* (EGFP) under the control of the CMV promoter[55] by In-Fusion (Takara) using EcoRI restriction sites.

To generate the donor plasmids for in vivo experiments in the mouse retina (Fig. 2a and Supplementary Figs. 3 and 5), four constructs were produced (Supplementary Fig. 2b). The m*Rho* and scramble gRNA cassettes were cloned in the pAAV2.1-VMD2-EGFP[59] backbone by In-Fusion (Takara) using an AflII restriction site. DsRed donor DNAs for m*Rho* were cloned by In-Fusion (Takara) using AflII sites. Schematic representations of these plasmids (p1135: kozak-DsRed + gRNA, p1116: kozak-DsRed + scramble, p1453: IRES-DsRed + gRNA and p1454: IRES-DsRed + scramble) are depicted in Supplementary Fig. 2b.

Similarly, four constructs were generated (Supplementary Fig. 2c) for in vivo experiments in the pig retina (Fig. 2b and Supplementary Fig. 4). For plasmids p1126 (kozak-DsRed + gRNA) and p1118 (kozak-DsRed + scramble), donor DNAs were cloned in a reverse complementary orientation due to cloning restrictions. For the same reason, in plasmid p1227 (IRES-DsRed + gRNA), the p*RHO*-gRNA expression cassette was cloned in a reverse complementary orientation. As a negative control for IRES-DsRed + gRNA, we used the plasmid p1048 (IRES-DsRed + scramble) generated for the mouse retina, which includes the same IRES-DsRed donor DNA and gRNA target sites which are not expected to be targeted by the scramble gRNA. Schematic representations of these plasmids are depicted in Supplementary Fig. 2c.

To generate the therapeutic donor plasmids for in vivo experiments in *Rho*^P23H−/+^ mice (Fig. 2c–e and Supplementary Fig. 5), the CDS for both h*RHO* and DsRed, including the ribosomal skipping sequence from Thosea asigna virus (2A) and IRES, as START site (Supplementary Fig. 2d), was generated by PCR amplification of a plasmid containing the h*RHO*-2A-DsRed CDS. PCR fragments were subcloned in PCR-Blunt II-TOPO (Invitrogen) before cloning in the EcoRI restriction sites of a pAAV2.1 plasmid by In-Fusion (Takara). After this, the m*Rho* gRNA cassette was cloned by In-Fusion (Takara) using an EcoRV restriction site to generate plasmid p1445 (IRES-h*RHO* + gRNA). To generate the control plasmid p1447 (IRES-h*RHO* + scramble), the m*Rho* gRNA expression cassette was substituted by a scramble gRNA expression cassette obtained from p1161 by SpeI + EcoRV restriction and ligation. Schematic representations of these plasmids are depicted in Supplementary Fig. 2d.

The pAAV2.1 plasmid bearing SpCas9 under the control of a synthesized HLP[29] and the pA sequence (p1139: HLP-SpCas9, Supplementary Fig. 6a) was generated by substituting the IRBP promoter with HLP, by In-Fusion cloning (Takara) using AflII and AgeI restriction sites.

To generate the donor plasmids for in vivo experiments in mouse liver (Fig. 3a, Supplementary Figs. 7 and 8 and Supplementary Table 5), kozak-DsRed donor DNA was cloned in PCR-Blunt II TOPO (Invitrogen) and then by In-Fusion (Takara) in a pAAV2.1 plasmid using EcoRI restriction sites. STOP codons and kozak, as well as the 5′ and 3′ mAlb gRNA target sites were added as PCR primer overhangs. Then, mAlb and scramble gRNA expression cassettes, previously subcloned in PCR-Blunt II-TOPO (Invitrogen), were cloned by In-Fusion (Takara) using an EcoRV restriction site to obtain plasmid p1160 (kozak-DsRed + gRNA) and p1161 (kozak-DsRed + scramble). In addition, DsRed CDS was cloned into a pAAV2.1 under the control of the TBG promoter using NotI and BglII restriction sites by conventional ligation to obtain plasmid p1444 (TBG-DsRed). Schematic representations of these plasmids are depicted in Supplementary Fig. 6b.

Plasmid p1160 was used for In-Fusion (Takara) mutagenesis in order to replace the kozak with the IRES sequence and thus to obtain plasmid p1224 (IRES-DsRed + gRNA). The scramble gRNA expression cassette was obtained from p1161 and cloned into p1224 using SpeI + EcoRV restriction digestion and conventional ligation, to generate plasmid p1448 (IRES-DsRed + scramble). Schematic representations of these plasmids are depicted in Supplementary Fig. 6b.

To generate the therapeutic donor plasmids for in vivo experiments in MPS VI mice (Fig. 3b–i), the human Arylsulfatase B (hARSB) CDS was generated by PCR amplification of the pAAV2.1_TBG_hARSB plasmid published in[60], which contains both the hARSB CDS and the pA, by adding the mAlb gRNA target sites, STOP codons and kozak as PCR primers overhangs. The PCR fragment was subcloned in reverse complementary orientation in PCR-Blunt II-TOPO (Invitrogen) before cloning in a pAAV2.1 backbone by In-Fusion (Takara) using EcoRI restriction sites. After this, the mAlb and scramble gRNA cassettes from PCR-Blunt II TOPO (Invitrogen) were cloned by In-Fusion (Takara) using an AflII restriction site to obtain plasmids p1239 (kozak-hARSB + gRNA) and p1240 (kozak-hARSB + scramble), respectively. To avoid deletions in the donor DNA, a stuffer DNA was added to plasmid p1239 between the hARSB CDS and the pA by In-Fusion (Takara) using AflII and HindIII restriction sites to generate plasmid p1336 (kozak-hARSB + Stuffer + gRNA). Schematic representations of these plasmids are depicted in Supplementary Fig. 6c.

**AAV vector production and characterization**. AAV vectors serotype 8 (AAV8) were produced by the TIGEM AAV Vector Core and InnovaVector srl by triple transfection of HEK293 cells followed by two rounds of $CsCl_2$ purification[61]. For each viral preparation, physical titers (genome copies/ml) were determined by averaging the titer achieved by dot-blot analysis and by PCR quantification using TaqMan (Applied Biosystems, Carlsbad, California, USA)[61].

**Culture and transfection of cell lines**. Hepa1–6 (ATCC, CRL-1830), Pk15 (ATCC, CCL33) and HEK293 (ATCC, CRL-1573) cells, were maintained in DMEM containing 10% fetal bovine serum (FBS) and 2 mM L-glutamine (Gibco, Thermo Fisher Scientific, Waltham, MA, USA). Pk15 cells medium was supplemented with one volume of M199 medium (Gibco).

To evaluate gRNA cutting efficiency (Supplementary Table 1), either mouse (Hepa1–6) or pig (Pk15) cell lines were transfected with a plasmid encoding SpCas9-EGFP and either the mRho-, mAlb-, pRHO-gRNA or the scramble-gRNA using the Effectene transfection reagent (QIAGEN, Milan, Italy) following the manufacturer's instructions.

For in vitro assessment of HITI in mRho, HEK293 cells were transfected using the calcium phosphate method with: (1) CMV-mRho P23H plasmid; (2) SpCas9-EGFP plasmid as reporter, under the control of the chicken beta actin hybrid (CBH) promoter (CBH-SpCas9-EGFP), and either the mRho- or the scramble-gRNA expression cassettes and (3) either kozak-DsRed or IRES-DsRed donor plasmids.

To evaluate expression of hRHO following HITI with therapeutic constructs, HEK293 cells were transfected with the CMV-mRho P23H template plasmid, the CBH-SpCas9-EGFP plasmid and a plasmid containing the hRHO-2A-DsRed donor with either the mRho- or the scramble-gRNAs.

**Animal models**. Mice were housed at the TIGEM animal house (Pozzuoli, Italy) and maintained under a 12 h light/dark cycle at 23 ± 1 °C and humidity of 50 ± 5% with food and water available ad libitum. Animals were raised in accordance with the Institutional Animal Care and Use Committee guidelines for the care and use of animals in research.

C57BL/6J mice were purchased from Envigo Italy SRL (Udine, Italy). The Rho-P23H knock-in[28] mice (referred as $Rho^{P23H-/+}$) were imported from The Jackson Laboratory (Stock No: 017628). Mice were maintained by crossing homozygous females and males. Experimental heterozygous animals were generated by crossing homozygous $Rho^{P23H-/+}$ with C57BL/6J mice. Genotype analysis was performed by PCR analysis on genomic DNA, using the following primers: Fwd: 5′-TGGAAGGTCAATGAGGCTCT-3′; Rev: 5′-GACCCCACAGAGACAAGCTC-3.

The electroretinographic amplitudes recorded from retinas of untreated mice (e.g., treated with negative control scramble vector) are similar to the amplitudes observed at p90 by Mao et al.[62] in the same animal model.

The MPS VI mice, a kind gift of Prof. C. Peters (Institute of Molecular Medicine and Cell Research, University of Freidburg, Germany) carry a targeted disruption of the mouse Arsb locus[63,64] and are made immune-tolerant to hARSB by transgenic insertion of the C91S hARSB mutant[65], resulting in the production of inactive hARSB. Mice were maintained as heterozygous and crossed to produce homozygous knockout experimental mice. Genotype analysis was performed by PCR analysis on genomic DNA. Three different PCRs were performed using the following primers: (1) Fwd: 5′-TGGGCAGACTAGGTCTGG-3′ and Rev: 5′-TGTCTTCCACATGTTGAAGC-3′ to discriminate affected (knockout) from not affected mice; (2) Fwd: 5′-TCTGGAGGCAACAACTGGC-3′ and Rev: 5′-CGCGTCACCTTAATATGCGA-3′ to discriminate wild-type from heterozygous mice; (3) Fwd: 5′-TTAAGAAGCTGATAAAATCTGCAACAC-3′ and Rev: 5′-AACAATCAAGGGTCCCCAAAC-3′ to check for the presence of the C91S hARSB transgene.

The Large White female pigs (imported from Azienda Agricola Pasotti, Imola, Italy) used in this study were registered as purebred in the LWHerd Book of the Italian National Pig Breeders' Association and were housed at the Centro di Biotecnologie A.O.R.N. Antonio Cardarelli (Naples, Italy) and maintained under a 12 h light/dark cycle.

**Subretinal injection of AAV vectors in mice and pigs**. Studies in animals were carried out in accordance with both the Association for Research in Vision and Ophthalmology Statement for the Use of Animals in Ophthalmic and Vision Research and with the Italian Ministry of Health regulation for animal procedures (Ministry of Health authorization number: 588/2019-PR and 147/2015-PR). Surgery was performed under general anesthesia, and all efforts were made to minimize animal suffering. Mice (C57BL/6J or $Rho^{P23H-/+}$) were anesthetized with an intraperitoneal injection of 10 μl/g of body weight of ketamine (10 mg/Kg) combined with medetomidine (1 mg/Kg), then AAV8 vectors were delivered subretinally via a trans-scleral trans-choroidal approach, as described by Liang et al.[66]. Eyes were injected with 0.7–1 μl of vector solution in the temporal-ventral side of the eye.

For proof-of-concept and rescue experiments in the mouse retina, $2.5 \times 10^9$ or $1.7 \times 10^9$ genome copies/eye of each vector (IRBP-SpCas9 and the donor DNA with either kozak or IRES and gRNA or scramble) were subretinally injected in 4-week-old C57BL/6J and 7-day old $Rho^{P23H-/+}$ mice, respectively.

For experiments in the pig retina, $2 \times 10^{11}$ genome copies/eye of each vector (IRBP-SpCas9 or the donor DNA with either kozak or IRES and gRNA or scramble) were injected in 3-month-old Large White pigs.

Delivery of AAV8 vectors to the pig retina was performed in the subretinal space in the nasal area through a transconjunctival scleral incision described in[67]. Pigs were anesthetized with an intramuscular injection of Zoletil (0.5 ml/Kg), ketamine (10 mg/Kg) and propofol (6 mg/Kg). Anesthesia was maintained during the procedure with sevoflurane (2%). Eyes were injected with two blebs of 100 μl of AAV8 vector solution. AAV8 vectors were used because of their high efficiency of photoreceptor transduction[67,68].

**Neonatal intravenous injection via the temporal vein**. Studies in animals were carried out in accordance with the Italian Ministry of Health regulation for animal procedures (Ministry of Health authorization number: 113/2015-PR and 352/2020). The injections were performed under general anesthesia, as described above. Mice (C57BL/6J or MPS VI) were injected at post-natal (p)1-p2 through the temporal vein following the protocol published by Gombash Lampe et al.[69]. For proof of concept experiments in C57BL/6J mice, $4 \times 10^{13}$ genome copies /Kg of each vector were injected (HLP-SpCas9 and the donor DNA with either kozak or IRES and gRNA or scramble; or TBG-DsRed). For rescue experiments in MPS VI mice, $6 \times 10^{13}$ genome copies/Kg of each vector were injected (HLP-SpCas9 or the donor DNA with either gRNA or scramble). The total volume of injection was 40 μl. AAV8 vectors were used because of their high liver transduction efficiency[70].

**Cytofluorimetric analysis**. Hepa1–6, Pk15 and HEK293 cells transfected as described above, were washed once with PBS, detached with trypsin 0.05% EDTA (Thermo Fisher Scientific), washed twice with PBS, and resuspended in sorting solution containing: PBS, 5% FBS and 2.5 mM EDTA. Flow cytometry analysis was performed by the TIGEM Fluorescence Cytometry Core using the BD FACS ARIA III (BD Biosciences, San Jose, CA, USA) equipped with BD FACS Diva software with appropriate excitation and detection settings for either EGFP (for Hepa1-6, Pk15 and HEK293) or EGFP/DsRed (for HEK293). Thresholds for fluorescence detection were set using untransfected cells, and a minimum of 10,000 cells/sample were analyzed. A minimum of 5000 EGFP+ or EGFP+/DsRed+ cells/sample were sorted and used for either DNA or RNA extraction.

**DNA extraction**. DNA extraction from either EGFP+ (for Hepa1-6, Pk15 and HEK293) or EGFP+/DsRed+ (for HEK293) cells, as well as from pig retina, was performed using a commercial lysis buffer (GeneArt™ Genomic Cleavage Detection

Kit, Invitrogen). For all experiments performed in mouse retina and liver, DNA extraction was performed using the DNeasy Blood & Tissue kit (QIAGEN).

**RNA extraction and hRHO expression.** Total RNA was extracted using the RNeasy MiniKit (QIAGEN) from both EGFP+/DsRed− and EGFP+/DsRed+ sorted HEK293 cells. RNA (5–15 ng) was used as a template for One-Step RT-qPCR (NEB, Massachusetts, USA) according to the manufacturer's instructions using the LightCycler 96 (Roche Molecular Systems, Inc.). Expression levels of hRHO were normalized vs. the corresponding housekeeping gene (ACTB). The relative quantification was done using the $2^{(-\Delta\Delta Ct)}$ method. The primers used for the real-time qPCR amplification are as follows: Fwd hRHO: 5′-ACTTCCGCTTCGGGGGAGAACCA-3′; Rev hRHO: 5′-ATTCCA-CACGAGCACTGCAGGC-3.

**Tracking of INDELs by decomposition and T7 Endonuclease I assay.** Fifty to 200 ng of DNA extracted from EGFP+ sorted Hepa1-6 or Pk15 cells, retina or liver were used for PCR amplification of the region which encompasses the SpCas9 target site in: (1) Rho exon 1 which is included both in the CMV-mRho-P23H plasmid or in the mouse or pig genomic loci, respectively and (2) Alb exon 2. The following primers were used: mRho-P23H-plasmid-INDELs Fwd: 5′-CCATG GTGATGCGGTTTTGG-3′; mRho-P23H-plasmid-INDELs Rev: 5′-GGTGAAG ACCACACCCATGA-3′; mRho-INDELs Fwd: 5′-CAGTGCCTGGAGTTG CGCTG-3′; mRho-INDELs Rev: 5′-GGGCCCAAAGACGAAGTAGCC-3′; pRHO-INDELs Fwd: 5′-AGGCCTCAGCAGCATCCTTG-3′; pRHO-INDELs Rev: 5′-GTGGTGGTGAAGCCTCCGAA-3′; mAlb-INDELs Fwd: 5′-ATTACGGTCTCAT AGGGCCTGC-3′; mAlb-INDELs Rev: 5′-GCACACATTTCTACTGGACAGCA-3′. PCR products were amplified and acquired using the T100 Thermal Cycler (Bio-Rad Laboratories, Inc) and GelDoc XR+ (Bio-Rad Laboratories, Inc), respectively.

PCR products were sequenced by Sanger sequencing and then analyzed by TIDE software (https://tide.deskgen.com) to assess INDEL frequency. For analysis, the sequence from each gRNA or scramble-treated cells, mouse and pig retina or mouse liver were matched to those from untransfected cells, un-injected retina (mouse), retinal pigmented epithelium (RPE) or PBS-injected liver, respectively. Mean INDEL frequencies for each sample were then calculated. INDEL frequencies with a p value either <0.01 or <0.001 were considered for retina and liver or cells, respectively.

Between 1 and 3 µl of the PCR products amplified from pig retina DNA (volume selected according to PCR efficiency), were used for T7 Endonuclease 1 (T7E1) assay using the GeneArt™ Genomic Cleavage Detection Kit (Invitrogen), following manufacturer's recommendations. Samples were run on a 2% agarose gel in order to detect DNA cleavage products resulting from INDEL presence.

**gRNA off-target site analysis.** The top 10 predicted off-target sites (for either mRho or mAlb gRNAs) were identified using the CRISPOR web tool (http://crispor.org), based on GRCm39/mm39 mouse genome reference, and sorted by CFD off-target scores. Equal amounts of genomic DNA extracted from 3 gRNA-treated retinas or livers and 2 scramble-treated retinas or liver were used to amplify 150 bp genomic regions flanking the on-target and off-targets for library construction. PCR products were quantified using Qubit High Sensitivity kit (Invitrogen) and then equal amounts of PCR products from each retina or liver sample were pooled together; for next-generation sequencing (NGS) library preparation, 10 ng of pooled PCR products were used as input for the NEBNext® Ultra™ II DNA Library Prep Kit for Illumina (New England Biolabs, Ipswich, MA, USA). The quality of libraries was assessed by using screen tape High sensitivity DNA D1000 (Agilent Technologies, Santa Clara, CA, USA), and quantified by using the Qubit 4 Fluorometer (Thermo Fisher Scientific). Deep sequencing was performed through MiSeq system using a PE 150 cycles strategy on a Nano V2 flow cell (Illumina, San Diego, USA).

Illumina Miseq base call (BCL) files were converted in fastq file through bcl2fastq (version v2.20.0.422) and then merged and processed by CRISPRessoV2[71], using the on-target and off-targets sequences as reference for analysis. A 17-nucleotide window (upstream and downstream of the cleavage site) was considered for evaluation. NGS analysis was performed by Next Generation Diagnostic srl.

**Analysis of large insertions and deletions.** To identify large chromosomal rearrangements induced by SpCas9 at the genomic target sites (either mRho and mAlb), 9 kb fragments upstream and downstream of the target site as well as the on-target site were amplified, using either the GoTaq Long PCR Master Mix (Promega, Madison, WI, USA) or PrimerSTAR GXL DNA polymerase (Takara). The following primers were used: mRho insertion Fwd: 5′-GAGGGCCC-CAATTTTTATGT-3′; mRho insertion Rev: 5′-GGAAGTTGATGGGGAAGC-3′; mRho deletion (upstream) Fwd: 5′-CCCCTGGTGCTGTCCAATAG-3′; mRho deletion (upstream) Rev: 5′-AGCACGATGAGCAGGAACAT-3′; mRho deletion (downstream) Fwd: 5′-CCCTTCTCCAACGTCACAGG-3′; mRho deletion (downstream) Rev: 5′-CAAGTTCCATCCCCAGGACC-3′; mAlb insertion Fwd: 5′-GCATATTACAGTTAGTTGTCTTCATCA-3′; mAlb insertion Rev: 5′-TGGGCTTTCAGCATTATAACTT-3′; mAlb deletion (upstream) Fwd: 5′-CTCCTCCCACACAACCATTT-3′; mAlb deletion (upstream); Rev: 5′-

TTAAGTGGGCTTTCAGCATT-3′; mAlb deletion (downstream) Fwd: 5′-ATCTTTAAATATGTTGTGTGGTTTTTCTCT-3′; mAlb deletion (downstream) Rev: 5′-ACAAGTAAAAGACAAGTCAGGAGATCTTT-3′.

**HITI efficiency by NGS analysis.** To quantify both SpCas9 cleavage and HITI outcomes in the mouse and pig retina, sequencing analysis by NGS at the targeted locus from two gRNA-treated retinas was performed. Using DNA as a template, two pairs of primers (with Fwd primer in common between the wild-type and the integrated allele) with similar efficiencies were designed and similar size region of both wild-type allele (mRho or pRHO) and hybrid allele including the integrated donor DNA (mRho + DsRed or pRHO + DsRed) were PCR-amplified. PCR products were co-purified and sequenced using the Illumina MiSeq platform and processed by CRISPRessoV2[71], using the either wild-type or integrated donor sequences as reference for analysis. A 17-nucleotide window (upstream and downstream the cleavage site) was considered for evaluation. The following primers were used: wild-type/integrated mRho Fwd: 5′-CAACGTCACAGGCGTG GT-3′; wild-type mRho Rev: 5′-TTCTTGTGCTGTACGGTGACGTAGAG-3; integrated mRho Rev: 5′-GGCTTGATGACGTTCTCAGTG-3′; wild-type/integrated pRHO Fwd: 5′-TACGTGCCTTTCTCCAACAA-3′; wild-type pRHO Rev: 5′- AGAGCGTGAGGAAGTTGATG-3′; integrated pRHO Rev: 5′-ATGAAG GGCTTGATGACGTT-3′. NGS analysis was performed by Next Generation Diagnostic srl.

**HITI junction characterization.** DNA extracted from one gRNA-treated retina or liver, harvested 30 days after injection, was used for PCR amplification of HITI junctions. Both 5′ and 3′ junctions of integration were amplified. The following primers were used: mRho HITI 5′ junction Fwd: 5′-TTGGTCTCTG TCTACGAAGAGCC-3′; mRho HITI 5′ junction Rev: 5′-GGCTTGATGAC GTTCTCAGTGC-3′; mRho HITI 3′ junction Fwd: 5′-CGACCTGCA GAAGCTTGGATCT-3′; mRho HITI 3′ junction Rev: 5′-GGGCCCAAA GACGAAGTAGCC-3′; mAlb HITI 5′ junction Fwd: 5′-GCCTGCTCGAC CATGCTATACT-3′; mAlb HITI 5′ junction Rev: 5′-CCTTGGAGCCGTACTG GAACTG-3′; mAlb HITI 3′ junction Fwd: 5′-CGACCTGCAGAAGCTTGGA TCT-3′; mAlb HITI 3′ junction Rev: 5′-TCTCTGGCTGCCACATTGCT-3′.

PCR products were cloned into PCR-Blunt II-TOPO (Invitrogen) and single clones were sequenced to confirm the identity of the PCR products before NGS analysis.

For NGS library preparation, 10 ng of DNA from the PCR products were used as input for the NEBNext® Ultra™ II DNA Library Prep Kit for Illumina (New England Biolabs). The quality of the libraries was assessed by using a Bioanalyzer DNA High Sensitivity chip (Agilent Technologies, Santa Clara, CA, USA), and quantified by Qubit (Invitrogen). The libraries were sequenced on a MiSeq sequencing system using a v3, 600 cycle flow cell (Illumina). The Illumina base call raw data were converted in fastq files through the bcl2fastq software (version v2.20.0.422, Illumina). The sequence reads were trimmed using the Cutadapt software (version 1.9) to reduce the reads length from 600 to 350 bp and then aligned with their respective reference sequence using the BWA-SW software (version 0.7.10, MIT, Boston, MA, USA). The total number of INDELs and specific nucleotide counts contained in the aligned BAM files were estimated with the deepSNV software package (version 1.30.0, Bioconductor). The length of INDELs was obtained from BAM CIGAR strings through an ad-hoc R algorithm. INDELs with a frequency ≥ than 0.1% (retina) or 0.5% (liver) were taken into account.

A 17-nucleotide window (upstream and downstream the cleavage site) was considered for evaluation. For the retina, the total reads were between 3059512-3082469 and 1401931-1419062 for the 5′ and 3′ junction, respectively. The reads containing the 40 and 48 bp insertions have been isolated using samtools (version 1.3)[72] and uploaded to IGV (version 2.4.17). The sequences identified by IGV have been quantified with an ad-hoc R script[72,73].

For liver, the total reads were between 117808-121889 and 418100-481800 for the 5′ and 3′ junction. NGS analysis was performed by Next Generation Diagnostic srl.

**On-target and off-target HITI characterization in liver.** SureSelect custom probes (Agilent Technologies) designed to enrich the specific regions of interest (the donor DNA, the mAlb locus and the AAV sequences) were designed under the supervision of Agilent technical support. Libraries were generated following the manufacturer's instructions for SureSelect^QXT Target Enrichment for Illumina Multiplexed Sequencing (Protocol Version E0, April 2018). In total, 50 ng of genomic DNA were fragmented, and adapters were added in a single enzymatic step. The adapter-tagged DNA library was purified and amplified. Next, 750 ng of each library were hybridized using the SureSelect^QXT capture library (Agilent Technologies). The resulting libraries were recovered using streptavidin magnetics beads, and a post-capture PCR amplification was carried out. Final libraries were quantified by Qubit (Invitrogen) and the libraries quality was assessed on a Bioanalyzer High Sensitivity DNA chip (Agilent Technologies). A first library concentrated 1.6 pM was used to carry out cluster generation and sequencing on a NextSeq-High Output Flow Cell and a second library concentrated 250 pM was run on NovaSeq6000 S1 flow cell (2 × 150 cycles) with 5% of phiX. Reads were merged and aligned to the mouse genome using the BWA-MEM software (Illumina).

To detect reads containing the donor DNA, we designed an ad-hoc procedure exploiting BWA, bedtools, picard and Bash/AWK commands for Linux[74,75]. Reads were aligned against the insert sequence using BWA parameters to achieve the highest possible sequence identity (BWA-MEM -O 60 -B 50 -E 10 -L 100 -c 2 -U 50 -d 200 -w 1). Output files in SAM format were parsed and only those reads resulting to have a partial alignment were kept because they provide information on the presence of the donor DNA. Such reads were further separated to obtain two different FASTA files: one with sequences containing the insert and the second with sequences which do not contain the insert. We set a threshold to only keep sequences longer than 22 nt (first FASTA) and 30 nt (second FASTA). These files were respectively realigned against the AAV genome and the mm10 genome. The resulting SAM files were further parsed to obtain the frequency of alignment per chromosome and to verify the part of the AAV genome that was integrated.

To detect the frequency of reads containing the donor DNA in the mAlb target site we used the following procedure: first the reads were aligned against the insert (which contains the donor DNA) using BWA to get as high as possible sequence identity (BWA-MEM -O 60 -B 50 -E 10 -L 100 -c 2 -U 50 -d 200 -w 1). The resulting alignment output file in SAM format was parsed with Linux commands to only extract reads that align with the first 10 nt of the insert. Then, using the CIGAR string, only reads containing a partial match were kept. Finally read identifiers were collected. A second alignment with the same BWA parameters, was performed against the entire mAlb gene. The read identifiers previously collected were used to filter this alignment. Hence, all the reads containing the insert and aligning against the mAlb gene were obtained. Then, to count the frequency of reads mapped in the cleavage site, reads falling in the region of cleavage were filtered using AWK commands with the SAM file and counted using Linux commands. A total of about 300,000 reads covering the target site in gRNA-treated livers and around 80,000 in scramble-treated were identified.

**Retinal cryosections and fluorescence imaging**. To evaluate DsRed expression in retinal histological sections after HITI, mice and pigs were sacrificed, and eyes were fixed in 4% PFA overnight and infiltrated with 30% sucrose overnight; the cornea and the lens were then dissected, and the eyecups were embedded in optimal cutting temperature compound (O.C.T. matrix; Kaltek, Padua, Italy). Ten-micrometer-thick serial retinal cryosections were cut along the horizontal meridian, progressively distributed on slides, and mounted with Vectashield with DAPI (Vector Lab). Cryosections were analyzed under a confocal LSM-700 microscope (Carl Zeiss), using appropriate excitation and detection settings for DsRed and DAPI. For assessment of HITI efficiency in mouse and pig retinal cryosections following AAV administration, the highest transduced area of two-three sections/eye (except for one eye in the IRES-gRNA group in Fig. 2a where only one slice was quantified) were selected and acquired at ×40 magnification and then analyzed by two independent observers using the ImageJ (Fiji) software (http://rsbweb.nih.gov/ij/). A minimum of 200 photoreceptors, identified by DAPI staining, were counted for each image. Photoreceptors with signal compatible with DsRed expression were unequivocally identified based on their shape. The analysis was made on the maximal projection of the z-stacks. When the nuclei shape was not well defined, single z-stacks were analyzed. The mean value for each eye was then calculated.

**Electroretinography**. The $Rho^{P23H-/+}$ mice were dark-adapted for 3 h. Mice were anesthetized and positioned in a stereotaxic apparatus, under dim red light. Pupils were dilated with a drop of 0.5% tropicamide (Visufarma, Rome, Italy) and body temperature was maintained at 37.5 °C. Light flashes were generated by a Ganzfeld stimulator (CSO, Costruzione Strumenti Oftalmici, Florence, Italy). Electrophysiological signals were recorded through gold-plate electrodes inserted under the lower eyelids in contact with the cornea. Electrodes in each eye were referred to a needle electrode inserted subcutaneously at the level of the corresponding frontal region. The different electrodes were connected to a two-channel amplifier. After completion of responses obtained in dark-adapted conditions (scotopic), the recording session continued with the purpose of dissecting the cone pathway mediating the light response (photopic). To minimize the noise, different responses evoked by light were averaged for each luminance step.

Maximal scotopic response of rods and cones was measured in dark conditions (scotopic) with two flashes of 0.7 Hz and a light intensity of 20 candela-seconds per meter squared (cd s/m²), photopic cone responses were isolated in light conditions with a continuous background white light of 50 cd s/m², with 10 flashes of 0.7 Hz and a light intensity of 20 cd s/m².

**Optomotry**. The visual acuity of $Rho^{P23H-/+}$ mice was measured by the optomotor system (OptoMotry; www.cerebralmechanics.com). The mouse was positioned on a pedestal located in the center of a chamber consisting of four LCD monitors inwards facing. After a few minutes, necessary to adapt the mouse to the new environment, the test begins; a pattern of sinus stripes rotating clockwise and anti-clockwise appears on the monitor as determined randomly by the OptoMotryTM software (version VR 1.4.0). The ability to discriminate the stimulus is considered positive (correct response) when the mouse moves the head in the same direction of the gratings rotation, having a constant contrast of 100% and increasing spatial frequencies (cycles/degree).

**Evaluation of retinal outer nuclear layer thickness**. To evaluate retinal ONL thickness on retinal histological sections, $Rho^{P23H-/+}$ mouse eyes were fixed overnight in Davidson's fixative (deionized water, 10% acetic acid, 20% formalin, 35% ethanol), dehydrated by serial passages in ethanol and then embedded in paraffin using Excelsior AS (ASHI, Italy) by the Advanced Histology Facility (Tigem, Italy). Ten-µm-thick sections were cut along the horizontal meridian, progressively distributed on slides and stained with Harris Hematoxylin and Eosin (Sigma-Aldrich). Then, the sections were analyzed under the microscope (Leica Microsystems GmbH DM-5000, Wetzlar, Germany) and acquired at ×20 magnification. For each eye, three images from the temporal injected side of different depth (ventral, central and dorsal side) representative of the whole eye were used for the analysis. Three measurements of the ONL thickness were taken for each image using the "freehand line" tool of the ImageJ (Fiji) software. This was done blind to the treatment group.

**Liver fluorescence imaging**. To evaluate DsRed expression in liver, C57BL/6J mice were injected at p2, and sacrificed at p15, p30 and p90 by anesthesia overdose followed by cardiac perfusion of PBS. Liver was harvested, and a small piece of each lobe was dissected, fixed with 4% PFA overnight, infiltrated with 15% sucrose over the course of a day and 30% sucrose overnight before being included in O.C.T. matrix (Kaltek) for cryo-sectioning. Five-µm-thick liver cryosections were cut, distributed on slides and mounted with Vectashield with DAPI (Vector Lab). Cryosections were analyzed under a confocal LSM-700 microscope (Carl Zeiss), using appropriate excitation and detection settings for DsRed and DAPI. For assessment of HITI efficiency in mouse liver cryosections, three-four images of each liver were acquired at ×20 magnification and then analyzed using the ImageJ (Fiji) software (http://rsbweb.nih.gov/ij/). A minimum of 1000 hepatocytes, identified by DAPI staining of the nucleus, were counted for each image. The mean value for each liver was then calculated. Hepatocytes with signal compatible with DsRed expression were unequivocally identified based on their shape.

**Serum ARSB enzymatic activity measurement**. Blood was collected at p30, p150 and p360 from MPS VI treated and control mice via eye bleeding and centrifuged at $16,000 \times g$ in a microcentrifuge (Heraeus Fresco 21; Thermo Scientific, Waltham, MA) for 10 min at 4 °C to obtain serum. Serum ARSB activity was measured by an immune capture assay based on the use of a specific custom-made anti-hARSB polyclonal antibody (Covalab, Villeurbanne, France). Briefly, 96-well plates (Nunc Immuno™ Micro-well, Sigma-Aldrich St. Louis, Missouri, USA) were coated with 5 µg/ml of anti-hARSB antibody in 0.1 M NaHCO₃ (100 µl/well) and incubated overnight at 4 °C. The following day, plates were blocked with 1% milk 0.25 M NaCl/ 0.02 M Tris pH 7.0; after 2 h of incubation, 50 µl of standard and unknown samples (diluted 1:10) were added to each well. Plates were incubated at 4 °C overnight. The following day, 100 µl 5 mM 4-methylumbelliferylsulfate potassium salt (4-MUS; M-760-5, GoldBio, St Louis, Missouri, USA) substrate were added to each well and then incubated at 37 °C for 4 h. The reaction was stopped by the addition of 100 µl/well of 0.2 M glycine. Plates were shaken for 10 min at room temperature and fluorescence was read (excitation of 365 nm/emission of 460 nm) on a multiplate fluorimeter (Infinite F200; TECAN, Männedorf, Switzerland). Serum ARSB activity is expressed as pg/ml of serum, using a standard curve of recombinant hARSB (Naglazyme; BioMarin Europe, UK). Since the current anti-hARSB antibody recognizes mouse ARSB (mARSB) less efficiently than others used previously, to compare levels of recombinant hARSB to the wild-type endogenous mARSB (to calculate the % of normal obtained following AAV-HITI), serum ARSB activity in samples expressing hARSB was determined based on a historical standard curve of recombinant hARSB used to calculate mARSB in normal control mice when an anti-hARSB antibody able to efficiently recognize also mARSB was available[32].

**Liver ARSB activity measurement**. Liver was homogenized in water and protein concentrations were determined by the BCA protein assay reagent (Pierce Protein Research Products; Thermo Fisher Scientific, Rockford, IL). Briefly, 20 µg of protein were incubated with 40 µl of 12.5 mM 4-methylumbelliferylsulfate substrate (4-MUS; M-760-5, GoldBio) for 3 h at 37 °C in the presence of 40 µl of 0.75 mM silver nitrate (Carlo Erba, Milan, Italy), which is known to inhibit the activity of other sulfatases. The reaction was stopped by adding 200 µl of carbonate glycine buffer and the fluorescence of the 4-methylumbelliferone produced was measured on a multiplate reader (Infinite F200, TECAN) at 365 nm (excitation) and 460 nm (emission). Enzyme activities were calculated using a standard curve of the fluorogenic 4-methylumbelliferone product (Sigma-Aldrich). Activity is expressed as nmol/mg/h.

**Quantitative analysis of GAG levels in urine and tissues**. Urine samples were collected over 24 h using metabolic cages at p30, p150 and p360 from MPS VI treated and from control mice. Samples were briefly centrifuged to remove debris and diluted 1:50 in water to measure GAG content. Fifty µl of diluted urine or 250 µg of protein lysate were then used for GAGs evaluation by reaction with Dimethyl methylene Blue (DMB, 341088, Sigma-Aldrich)[76]. Absorbance was read at 520 nm using a multiplate reader (Infinite F200, TECAN). GAG concentrations were determined based on a dermatan sulfate standard curve (C3788, Sigma-Aldrich). Tissue GAGs are expressed as µg of GAG per milligram of protein (µg/

mg of protein). Urinary GAGs were normalized to creatinine content, which was measured with a creatinine assay kit (Quidel, San Diego, USA) and expressed as µg of GAGs/µmol of creatinine.

**Bone histology**. Formalin-fixed bones were decalcified in 10% EDTA (Sigma-Aldrich, E5134), dehydrated, embedded in paraffin using Excelsior AS (ASHI, Italy) by the Advanced Histology Facility (Tigem, Italy), and sectioned in 7 µm-thick sections on a microtome. Sections were stained with Harris Haematoxylin and Eosin Y (Sigma-Aldrich) with standard procedures, including a differentiation step in Ammonia solution (Fisher Scientific, Waltham, MA, USA). Sections were analyzed under a light microscope (Leica Microsystems GmbH DM-5000) and acquired at ×40 magnification by a blind operator. Images for quantification of osteocyte vacuoles were acquired using the Axio Scan Z1 (Carl Zeiss). Measurements were performed using the "freehand" selection tool of the ImageJ (Fiji) software. Two or three different regions were evaluated for each sample; a minimum number of 30 vacuoles was measured for each sample. This was done blind to the treatment group.

**Serum albumin quantification**. Blood was collected at p360 from treated and control mice via eye bleeding and centrifuged as described above. Serum samples were diluted 1:30,000 and analyzed with a mouse albumin ELISA kit (Abcam, 108791, Cambridge, UK) following manufacturer's instructions. Serum albumin was expressed as mg of albumin/ml of serum.

**Prediction of genotype frequency at the m*Rho* locus**. The Hardy-Weinberg equation was used to predict the frequency of genotype outcomes following AAV-HITI in $Rho^{P23H-/+}$ mice. The alleles frequencies measured in gRNA-treated C57BL/6J mice (Supplementary Fig. 3d) were used for the calculation using the "Hardy-Weinberg equilibrium calculator" tool (https://scienceprimer.com/hardy-weinberg-equilibrium-calculator) under the assumption that in $Rho^{P23H-/+}$ mice the frequency of wild-type alleles is half (43%) of that found in C57BL/6J mice (86%, Supplementary Fig. 3d) and thus the frequency of the P23H allele is also predicted to be 43%.

**Statistical analysis**. Data are presented as either median (for non-parametric data) or mean ± SEM, which have been calculated using the number (*n*) of independent in vitro experiments or tissues, i.e., eyes or livers (not replicate measurements of the same sample). Statistical *p* values <0.05 were considered significant. The normality assumption was verified using the Shapiro–Wilk test. Data were analyzed by either the Student's *t* test or one/two-way ANOVA. When data were not normally distributed (Shapiro–Wilk test *p* value <0.05) either the Wilcoxon rank sum test or the Kruskal–Wallis rank sum test (non-parametric tests) were used. Pairwise comparisons between group levels with corrections for multiple testing were performed to determine whether the difference between specific pairs of groups is statistically significant.

Statistical comparisons were made as follows:

Figure 2. (a) DsRed+/ total photoreceptors (%). The unpaired *t*-test [statistics: $t(9.45) = 0.31$; effect size: 0.166] was used to compare the mean of two independent groups; *p* value = 0.7600. (b) DsRed+/ total photoreceptors (%). The unpaired *t*-test [statistics: $t(2.49) = 3.23$; effect size: 2.64] was used to compare the mean of two independent groups; *p* value = 0.0620. (c) Electroretinogram A and B-wave amplitude (µV). The statistical tests used to compare gRNA and scramble groups at different lux stimuli are as follows: A-wave: unpaired *t*-test [statistics: $t(7.34) = 2.47$; effect size: 1.377] (lux 20) $p = 0.0413$; unpaired *t*-test [statistics: $t(7.34) = 2.47$; effect size: 1.377] (lux 10) $p = 0.0413$; unpaired *t*-test [statistics: $t(13.12) = 1.22$; effect size: 0.615] (lux 1) $p = 0.2430$; unpaired *t*-test [statistics: $t(9.01) = -0.18$; effect size: -0.097] (lux 0.1) $p = 0.8610$; unpaired *t*-test [statistics: $t(14) = 0.65$; effect size: 0.318] (lux 0.01) $p = 0.5240$; unpaired *t*-test [statistics: $t(11.12) = 1.26$; effect size: 0.655] (photopic) $p = 0.2340$.

B-wave: Wilcoxon rank sum test [statistics: $W = 53.5$; effect size: 0.583] (lux 20) $p = 0.0227$; unpaired *t*-test [statistics: $t(11.31) = 1.06$; effect size: 0.550] (lux 10) $p = 0.3100$; unpaired *t*-test [statistics: $t(13.46) = 1.57$; effect size: 0.746] (lux 1) $p = 0.1390$; unpaired *t*-test [statistics: $t(13.46) = 1.57$; effect size: 0.746] (lux 0.1) $p = 0.1390$; unpaired *t*-test [statistics: $t(13.46) = 1.57$; effect size: 0.746] (lux 0.01) $p = 0.1390$; unpaired *t*-test [statistics: $t(13.92) = 2.03$; effect size: 0.996] (photopic) $p = 0.0622$. (d) Visual acuity (cycles/degree). The Wilcoxon rank sum test [statistics: $W = 36$; effect size: 0.832] was used to compare gRNAs and scramble groups; *p* value = 0.0022. (e) ONL thickness (µm). The unpaired *t*-test [statistics: $t(9.54) = 2.8$; effect size: 1.62] was used to compare the mean of two independent groups (whole); the *p* value is 0.0200.

Figure 3. (a) DsRed+/ total hepatocytes (%). The Kruskal–Wallis test [statistics: $\chi^2(5) = 18.95$; effect size: 0.698] followed by the Conover's all-pairs rank comparison test was used to perform multiple pairwise comparisons between groups. Fdr-adjustment was applied. The Kruskal–Wallis test *p* value = 0.0020. The Conover's test *p* values are as follows: TBG-DsRed p15 vs. gRNA p15 = 0.0005; TBG-DsRed p15 vs. gRNA p30 = 0.0005; TBG-DsRed p15 vs. gRNA p90 = 1.67e−05; TBG-DsRed p15 vs. TBG-DsRed p30 = 0.1896; TBG-DsRed p15 vs. TBG-DsRed p90 = 0.0014; TBG-DsRed p30 vs. TBG-DsRed p90 = 0.0454; TBG-DsRed p30 vs. gRNA p15 = 0.0085; TBG-DsRed p30 vs. gRNA p30 = 0.0104; TBG-DsRed p30 vs. gRNA p90 = 0.0005; TBG-DsRed p90 vs. gRNA p15 = 0.4093; TBG-DsRed p90 vs. gRNA p30 = 0.4683; TBG-DsRed p90 vs. gRNA p90 = 0.1059; gRNA p15 vs. gRNA

p30 = 0.8857; gRNA p15 vs. gRNA p90 = 0.5900; gRNA p30 vs. gRNA p90 = 0.5133. (b) Serum ARSB activity. The one-way ANOVA test [statistics: $F(2,14) = 0.47$; effect size: 0.063] was used to compare gRNA values at different timepoints. The ANOVA *p* value is 0.6320. Tukey adjustment was applied. The Wilcoxon rank sum test was used to compare gRNA and scramble groups at each timepoint; the *p* values are as follows: p30 = 0.0256; p150 = 0.0250; p360 = 0.0325 (statistics: $W = 18$, 18 and 15 for p30, p150 and p360, respectively; effect size: = 0.79, 0.79 and 0.81 for p30, p150 and p360, respectively). (c) Liver ARSB activity. The one-way ANOVA test [statistics: $F(2,12) = 11.2$; effect size: 0.65] followed by the Tukey post hoc test was used to perform multiple pairwise comparisons between groups. Tukey adjustment was applied. The ANOVA test *p* value is 0.0020. The Tukey post hoc test *p* values are as follows: NR vs. AF = 0.0017; gRNA vs. AF = 0.0059; gRNA vs. NR = 0.9690. (d) Urinary GAGs. The one-way ANOVA test [statistics: $F(8,37) = 14.8$; effect size: 0.76] followed by Tukey post hoc test was used to perform multiple pairwise comparisons between groups. Tukey adjustment was applied. The ANOVA *p* value is 1.96e−09. The Tukey post hoc test *p* values are as follows: at p30, NR vs. scramble = 0.0037; gRNA vs. scramble = 0.9490; NR vs. gRNA = 0.0290; at p150, NR vs. scramble = 4.85e−05; gRNA vs. scramble = 0.0256; NR vs. gRNA = 0.2320; at p360, NR vs. scramble = 2.50e−06; gRNA vs. scramble = 0.0043; NR vs. gRNA = 0.2150. (e) Area of osteocyte vacuoles. The one-way ANOVA test [statistics: $F(2,10) = 111.98$; effect size: 0.96] followed by the Tukey post hoc test was used to perform multiple pairwise comparisons between groups. Tukey adjustment was applied. The ANOVA test *p* value is 1.43e−07. The Tukey post hoc test *p* values are as follows: NR vs. AF = 2.00e−07; gRNA vs. AF = 9.00e−07; gRNA vs. NR = 0.284. (f) Liver GAGs. The Kruskal–Wallis test [statistics: $\chi^2(2) = 6.83$; effect size: 0.402] followed by the Conover's all-pairs rank comparison test was used to perform multiple pairwise comparisons between groups. Bonferroni adjustment was applied. The Kruskal–Wallis test is 0.0330. The Conover's test *p* values are as follows: NR vs. AF = 0.0190; gRNA vs. AF = 0.0630; gRNA vs. NR = 1. (g) Kidney GAGs. The Welch ANOVA test [statistics: $F(2,3.85) = 239.01$; effect size: 0.97] followed by the Games Howell post hoc test was used to perform multiple pairwise comparisons between groups. Tukey adjustment was applied. The ANOVA test *p* value is <0.0001. The Games Howell post hoc test *p* values are as follows: NR vs. AF = 0.0020; gRNA vs. AF = 0.0020; gRNA vs. NR = 0.9990. (h) Spleen GAGs. The one-way ANOVA test [statistics: $F(2,12) = 1410.85$; effect size: 1] followed by the Tukey post hoc test was used to perform multiple pairwise comparisons between groups. Tukey adjustment was applied. The ANOVA test *p* value is <0.0001. The Tukey post hoc test *p* values are as follows: NR vs. AF = < 0.0001; gRNA vs. AF = < 0.0001; gRNA vs. NR = 0.0663. (i) Serum albumin. The one-way ANOVA test [statistics: $F(2,17) = 2.46$; effect size: 0.22] followed by the Tukey post hoc test was used to perform multiple pairwise comparisons between groups. Tukey adjustment was applied. The ANOVA test *p* value is 0.1200.

**Reporting summary**. Further information on research design is available in the Nature Research Reporting Summary linked to this article.

## Data availability

Sequencing data are deposited on GEO database (https://www.ncbi.nlm.nih.gov/geo/query/acc.cgi). Specifically, data relative to HITI junction characterization are under the accession code GSE158771; data relative to HITI on- and off-target in liver are under the accession code GSE158759; data relative to SpCas9 gRNA off-targets are under the accession code GSE180117; data relative to HITI efficiency in the retina are under the accession code GSE180875. All data supporting the findings described in this manuscript are available in the article and in the Supplementary Information, and from the corresponding author upon reasonable request. Source data are provided with this paper.

## Code availability

The in-house script[77] to detect and quantify HITI on- and off-target in liver is publicly available at the following links: https://github.com/frankMusacchia/HITI_OffTargetDetection. Custom scripts used for other analyses are under IP evaluation, can be provided upon request (service@ngdx.eu) according to Next Generation Diagnostic srl MTA policy, and will be shared within 1 business day from MTA subscription.

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

## Acknowledgements

We gratefully acknowledge Graciana Diez-Roux and Phoebe Ashley-Norman (TIGEM Scientific Office). Figure 1 was created with Biorender.com. This work was supported by the European Research Council Advanced grant "EYEGET" (grant #694323), the European Union Horizon 2020 grant "UPGRADE" (grant #825825) and the Foundation Fighting Blindness Individual Research Award (grant #TA-GT-0619-0762-FED) to A.A., the Fondazione Telethon Core Grant, the Armenise-Harvard Foundation Career Development Award, the European Research Council Starting grant "CellKarma" (grant #759154), and the Rita-Levi Montalcini program from MIUR to D.C.

## Author contributions

M.L. and A.A. conceived the study. P.T., R.F., M.L. and A.A. designed the experiments and wrote the original manuscript. P.T., M.L. and M.C. performed experiments in vitro and in the retina. R.F., M.L., M.D., F.E. and E.P. performed liver experiments. M.L., H.L. and F.E. generated plasmids for in vitro experiments and AAV production. E.M. performed electrophysiological analyses. R.M., C.I., S.R., E.M.S. and E.N. performed in vivo procedures. A.M., L.DF. and D.C. performed the NGS analysis for both retina and liver experiments. A.T., G.P., F.M. and V.N. designed and performed HITI on-target and off-target analysis for liver experiments. A.I. performed statistical analyses.

## Competing interests

M.L. and A.A are listed as inventors on the patent application WO2020/0793033 "Homology independent targeted integration (HITI) for gene correction in photoreceptors" related to this work. A.A. and D.C. are founders, shareholders, and consultants of InnovaVector srl and Next Generation Diagnostic srl, respectively. A.A. is founder, shareholder and consultant of AAVantgarde Bio. R.F. is a consultant of AAVantgarde Bio. The remaining authors declare no competing interests.
