## [Peer Review File · Nature Communications]

Reviewers' Comments:

Reviewer #1:

Remarks to the Author:

Summary:

In this manuscript, the authors use CRISPR/Cas9-mediated homology-independent targeted integration (HITI) in preclinical studies for two genetic diseases with different inheritance mechanisms. In RP4, an autosomal dominant disease they demonstrate a ~1-4.5% targeted integration in the retina around the site of injection as well as changes in electroretinograms and on histology. The studies also show targeted integration in the liver into the albumin locus for the treatment of MSPVI. Integration efficiency of the missing enzyme is ~2% which results in partial enzymatic correction and reduction in excretion of urine GAGs. The studies are ambitious as they are trying to characterize two different disease models. For RP4 the authors use pigs and mice to look at efficiency and perform some characterization of the integrations.

Notably, these are not the first proof-of-concept studies of HITI in animal models (see Suzuki, Nature 540, 144–149, 2016). Most importantly, the studies fail to address the biggest concern about using HITI for clinical applications: its safety and the potential for genotoxicity. The authors looked at captured sequences into non-targeted loci but did not evaluate off-target effects of the guides used (by deep sequencing of predicted off-target loci or unbiased analysis) and they did not examine the possibility of larger deletions or rearrangements of the targeted locus.

Major comments:

1. One of the major problems with the paper is that it is trying to condense two different stories leaving a lot of unanswered questions and the reader with the feeling that each model was not thoroughly/carefully evaluated.
2. The authors recognize that AAV or donor capture at non-targeted loci is an issue. Doesn't this mean that a cutting guide would be a better control than the scrambled guide used in all experiments? At least this gives you an idea of modification at off-target sites.
3. As a proof-of-concept of therapeutic efficacy there is limited phenotypic characterization. Is 1-4% integration enough to impact performance is a visible platform test (Morris water maze)? What about phenotypic features of MPSVI mice such as the skeletal manifestations.
4. The most important concern about the use of engineered nucleases is the possibility of cutting at unintended sites, as well as larger chromosomal alterations, HITI but cutting at two sides makes it even more relevant. You should look at off-target effects of the guides used (by deep sequencing of predicted off-target loci or unbiased analysis) and also did look at the possibility of larger deletions or rearrangements of the targeted locus.
5. In Fig 1, explain lack of colocalization, presence of red cells that are not green – quantify this phenomenon which is evident in the one image shown.
6. There are discrepancies in the % indels generated at the targeted loci that were not targeted and the % targeted loci in the different models and the authors try to explain this in the discussion. It would be helpful for the reader to see the activity of the guides in cells without the donor and not in vivo. This is to get a better idea of how efficient the guides are. Some of these efficiencies seem very low but it seems to change between models like pig vs mouse.
7. I'm concerned about using reporter proteins as the only way to quantify the efficiency of modification. Please use an orthogonal, genomic tool to do it such as ddPCR or NGS.
8. Cells exposed to your AAVs can have several different genotypes as outcomes. Given that some of them are actually deleterious, such as knocking out Rho, you should do single-cell genotyping to quantify the proportions of these different genotypes.
9. What is the rationale for the performance of Kozac vs IRES vectors or is it just variable transduction?
10. You use the T7E1 assay to demonstrate the specificity of Cas9 expression. This assay is not sensitive. Also, TIDE's limit of detection is 2%.
11. Discuss competing for therapeutic approaches more explicitly and discuss advantages and disadvantages
12. How do your electroretinogram studies compared to what has been published for these mice?

Minor comments:

1. Per figure 1 endogenous promoter for the target gene is not the same as the promoter that drives exogenous Cas9 – redraw figure
2. Provide adequate references in the introduction around mutation prevalence, e.g. 40% of RP, is AD? The next line should refer specifically to Americans of European origin, not all patients in the United States. When possible cite primary references, not online book chapters. Also please cite relevant references for your animal models.
3. Express enzyme activity as percent of Wild type
4. Discuss why GAG's are not improved by p30 despite having added the enzyme on day 1 of life. Shouldn't it be enough time?
5. Provide albumin levels for normal mice

Reviewer #2:

Remarks to the Author:

In this paper the authors use homology-independent targeted integration (HITI) to both disrupt a mutant gene whilst inserting a therapeutic gene into a targeted locus using appropriate guide RNA sequences. The analyses are both in vitro and in vivo and tested in two organ systems: retina and liver. This is a very nice approach and one that may have widespread applications in CRISPR biology.

The authors present a huge amount of information and despite rereading it in detail several times, I was unable to follow the detail of what they were doing or why. For instance, there was no explanation that I could see as to why Kozak-DsRed or IRES-DsRed were compared. There are far too many acronyms which makes reading the text very difficult. The data are presented without unoperated controls and so one cannot assess any potential therapeutic effect (in other words, the scrambled gRNA may simply be more toxic than the targeted gRNA sequences).

The combination of two organ systems is unusual and I wonder if it might be better to break this paper into two separate papers - one for eye and one for liver? In both cases the analyses require more depth. More detail on the ERG analyses (e.g. a wave, unoperated control) and better retinal histology might help in vivo studies and the liver work would be better assessed with a more thorough analysis of liver histology and enzymes. The results are far too focussed on technical detail that should be supplemental and so the bigger picture of the message is lost.

In brief, I have enjoyed reading the paper and this is a nice application of HITI which appears to work to some extent at least in vivo. I do not however feel qualified to give an opinion because I was unable to follow the detail of what was going on.

Point-by-point response

Reviewer #1 (Remarks to the Author):

Summary:

In this manuscript, the authors use CRISPR/Cas9-mediated homology-independent targeted integration (HITI) in preclinical studies for two genetic diseases with different inheritance mechanisms. In RP4, an autosomal dominant disease they demonstrate a ~1-4.5% targeted integration in the retina around the site of injection as well as changes in electroretinograms and on histology. The studies also show targeted integration in the liver into the albumin locus for the treatment of MSPVI. Integration efficiency of the missing enzyme is ~2% which results in partial enzymatic correction and reduction in excretion of urine GAGs. The studies are ambitious as they are trying to characterize two different disease models. For RP4 the authors use pigs and mice to look at efficiency and perform some characterization of the integrations.

Notably, these are not the first proof-of-concept studies of HITI in animal models (see Suzuki, Nature 540, 144–149, 2016). Most importantly, the studies fail to address the biggest concern about using HITI for clinical applications: its safety and the potential for genotoxicity. The authors looked at captured sequences into non-targeted loci but did not evaluate off-target effects of the guides used (by deep sequencing of predicted off-target loci or unbiased analysis) and they did not examine the possibility of larger deletions or rearrangements of the targeted locus.

We have performed deep sequencing analysis of predicted off-target as requested. In addition, we have evaluated rearrangements at the targeted locus, as also suggested. Please see the point-by-point answer below for more details.

Major comments:

1. One of the major problems with the paper is that it is trying to condense two different stories leaving a lot of unanswered questions and the reader with the feeling that each model was not thoroughly/carefully evaluated.

The reason we have applied HITI to both retina and liver is because this approach allows major limitations of these important targets of in vivo gene therapy to be overcome. We have included additional characterization results to assess HITI-mediated phenotypic improvements of both animal models of retinitis pigmentosa and mucopolysaccharidosis type VI, as requested. This makes the current characterization of HITI efficacy and safety particularly thorough. The description of the newly added data is detailed in points 3 and 12 of our answer to this Reviewer.

2. The authors recognize that AAV or donor capture at non-targeted loci is an issue.

Indeed, we have evaluated AAV or donor capture at non-targeted loci in liver and found that this was similar between animals receiving the therapeutic gRNA or the scramble gRNA, meaning that we

found no detectable capture specific to SpCas9 activity (Supplementary Results and Discussion, page 3, Fig. S8F and Table S4).

Doesn't this mean that a cutting guide would be a better control than the scrambled guide used in all experiments? At least this gives you an idea of modification at off-target sites.

As negative control we designed a scramble gRNA that does not align with murine or porcine genomic sequences. However, we have followed the Reviewer's suggestion and performed deep sequencing analysis of predicted off-target. See point 4 below for a description of the results.

3. As a proof-of-concept of therapeutic efficacy there is limited phenotypic characterization. Is 1-4% integration enough to impact performance in a visible platform test (Morris water maze)?

As an additional endpoint of visual function, we have used visual acuity measured by Optomotry (Prusky et al., Invest Ophthalmol Vis Sci. 2004 Dec;45(12):4611-6), a test that we and others routinely perform to measure visual function in rodents as an alternative to the more complex Morris water maze. One of the advantages of Optomotry is that it allows measurement of visual acuity from each of the two eyes of the same animal independently, unlike the Morris water maze (Douglas et al., Vis Neurosci. Sep-Oct 2005;22(5):677-84; Pearson et al., Nature. 2012 May 3;485(7396):99-103). Since our animals have the negative control treated eye contralateral to the gRNA-treated one, Optomotry is better suited than the Morris water maze for our experimental design. We show that visual acuity is significantly improved in the gRNA-treated eyes compared to the scramble-treated contralateral control eyes (Results section, page 6, Fig. 2D). This is not surprising given the improvement observed at the ERG level. Indeed, while the ERG response is mediated from the whole retina (including areas that are untreated), visual function can improve as result of few functioning photoreceptors. Also, as suggested by Reviewer 2, we have added ERG data relative to the A-wave of the experimental eyes which also confirm significant improvement (Results section, page 5-6, Fig. 2C).

What about phenotypic features of MPSVI mice such as the skeletal manifestations.

We have performed additional biochemical and histological rescue assessments in MPS VI mice, as suggested. We have measured GAG levels in liver, which is transduced by AAV, as well as spleen and kidney, which are not transduced by AAV but can be cross-corrected by recombinant ARSB secreted from transduced liver (Ferla et al., Hum Gene Ther. 2014 Jul;25(7):609-18). We show normalization of tissue GAG levels in gRNA-treated animals in Figure 3F-H. We have also assessed GAG storage by histological analysis and found rescue of GAG storage in liver and heart muscle and reduction of osteocyte vacuolization in the cortical bone of animals treated with gRNA compared to scramble-treated controls (Fig. 3E). For details, see the Results section, page 7.

4. The most important concern about the use of engineered nucleases is the possibility of cutting at unintended sites, as well as larger chromosomal alterations, HITI but cutting at two sides makes it even more relevant.

We agree with the Reviewer that one important concern regarding engineered nucleases is their potential off-target activity. We want to clarify that HITI produces a single cut at the endogenous

locus. The cutting at two sides only involves the donor DNA which is then integrated at the endogenous locus.

You should look at off-target effects of the guides used (by deep sequencing of predicted off-target loci or unbiased analysis)

As suggested by the Reviewer, we have used CRISPOR to predict off-targets of either mouse rhodopsin (mRho) or mouse albumin (mAlb) gRNAs. We have PCR-amplified the top 10 off-targets for each gRNA from either mouse retina or liver genomic DNAs. The PCR products for each eye or liver were pooled together and analyzed by deep sequencing using Illumina MiSeq system. Minimal INDEL frequency was found in 2 out of 10 of the predicted off-targets of mRho gRNA (Fig. S3B, Supplementary Results and Discussion section, page 1) suggesting that this effect of CRISPR/Cas9 is minimal. We did not detect differences in INDELS between gRNA and scramble samples at any of the predicted off-targets of mAlb gRNA (Fig. S8B, Supplementary Results and Discussion section, page 2).

and also did look at the possibility of larger deletions or rearrangements of the targeted locus.

To detect whether large insertions at the targeted locus occur, we set up a PCR amplification assay using primers flanking the target site. We did not detect any larger than expected products (Fig. S3C and Fig. S8C, insertion) suggesting that insertions are unlikely to occur. Similarly, to detect large deletions, we PCR amplified regions up to 9 kb long on each side of target site and did not detect any smaller product (Fig. S3C and Fig. S8C, deletion), suggesting that deletions within that range are also unlikely to occur. Please see Supplementary Results and Discussion section, page 1-2.

5. In Fig 2, explain lack of colocalization, presence of red cells that are not green – quantify this phenomenon which is evident in the one image shown.

Lack of co-localization in the image is presumably the result of strong red fluorescence which masks green fluorescence. We therefore substitute those images with FACS graph showing distribution of HEK293 cells based on EGFP and DsRed fluorescence, please see the current Supplementary Figure 1C.

6. There are discrepancies in the % indels generated at the targeted loci that were not targeted and the % targeted loci in the different models and the authors try to explain this in the discussion. It would be helpful for the reader to see the activity of the guides in cells without the donor and not in vivo. This is to get a better idea of how efficient the guides are. Some of these efficiencies seem very low but it seems to change between models like pig vs mouse.

According to the Reviewer suggestion, we have transfected either mouse (Hepa1-6) or pig (Pk15) cell lines with SpCas9 and the corresponding gRNAs, FAC-sorted cells that express SpCas9 and analyzed the predicted targets by TIDE. This was done in at least 3 independent experiments. The results of cleavage at each locus are shown in Supplementary Results and Discussion section, page 1 and Supplementary Figure 1A which shows that the overall in vitro cleavage activity is similar for the various gRNAs and ranges between 30 and 40%).

7. I'm concerned about using reporter proteins as the only way to quantify the efficiency of modification. Please use an orthogonal, genomic tool to do it such as ddPCR or NGS.

We agree with the Reviewer's concern. Actually, the data about integration of donor DNA at the mouse albumin locus are included in Supplementary Figure 8D which uses NGS to detect on- and off-target integrations of the donor DNA. While DsRed+ hepatocytes were about 2%, the reads at the albumin locus containing integration of the donor DNA were about 4%. This difference may be explained by bi-allelic editing potentially occurring in some hepatocytes (Supplementary Results and Discussion section, page 2).

For the retina, the same experiment done in liver (Fig. S8D) did not yield reliable results presumably due to the low number of edited photoreceptors. However, to answer the Reviewer request, we have PCR-amplified both the endogenous mouse and pig rhodopsin alleles (whether wild-type or carrying INDELS as result of SpCas9 cleavage without donor DNA integration) as well as the corresponding region including the breakpoint between the endogenous mouse/porcine rhodopsin and the integrated DsRed donor DNA. The resulting PCR products were analyzed by NGS and HITI efficiency observed by DsRed expression was confirmed, resulting in 5 and 4% of HITI in mouse and pig retina, and higher or similar INDEL frequency to that observed by TIDE analysis, respectively (Supplementary Results and Discussion section, page 1-2, Fig. S3A,D and Fig. S4B-C).

8. Cells exposed to your AAVs can have several different genotypes as outcomes. Given that some of them are actually deleterious, such as knocking out Rho, you should do single-cell genotyping to quantify the proportions of these different genotypes.

We agree with the Reviewer that different genotypes can arise from HITI at the rhodopsin locus. The experiment designed to answer the Reviewer's previous comment 7 also helps to assess the relative percentage of the various genotypes in a single treated retina (Fig. S3D). We trust that this will be an acceptable alternative to single cell sequencing. Setting up single-cell isolation from treated retinas and performing single-cell sequencing on hundreds of different photoreceptors would have necessitated a longer time and would have been challenging for our sequencing facility which is currently overloaded with COVID-19 patient sample analysis.

9. What is the rationale for the performance of Kozac vs IRES vectors or is it just variable transduction?

The two START sites of the donor DNA have different characteristics. On one hand we used kozak, which is the common signal for translation initiation by the ribosome. However, the presence of the kozak sequence from the endogenous gene could compete with the one from the donor DNA. On the other hand, we used a small synthetic IRES sequence which has been shown to efficiently recruit the ribosome (Venkatesan and Dasgupta, Mol Cell Biol. 2001 Apr;21(8):2826-37). However, since it would be integrated close to the endogenous gene translation start site, we did not know whether it would work efficiently. Our experiments demonstrate that both START signals mediate DsRed translation however with some differences. The better performance of the kozak constructs in vitro than in vivo could be due to the shorter distance between the kozak and the promoter in vitro, to the chromatin conformation facilitating ribosome recruitment by the IRES sequence or to photoreceptor-specific differences in protein translation. Additionally, the differences in efficiency of kozak-DsRed

and IRES-DsRed donors in pigs compared to mice could depend on the different site of insertion. This is now pointed out in the Discussion section (page 10).

10. You use the T7E1 assay to demonstrate the specificity of Cas9 expression. This assay is not sensitive. Also, TIDE's limit of detection is 2%.

We acknowledge the limited sensitivity of both T7E1 and TIDE. For this reason, we have performed NGS analysis of target sites in both liver and either mouse or pig retina in order to better quantify the INDEL efficiency. This shows an INDEL frequency that is either higher than (9% vs 3% in mouse retina) or similar to (19% vs 17% in pig retina) that of TIDE (Supplementary Results and Discussion section, page 1-2 and Fig. S3A, D and Fig. S4 B-C). In the liver, NGS analysis shows higher INDEL frequency relative to TIDE (21% vs 10%, Supplementary Results and Discussion section, page 2 and Fig. S8B).

11. Discuss competing for therapeutic approaches more explicitly and discuss advantages and disadvantages

We have expanded the Discussion section to include both the base and prime editing approaches and we have commented on their differences with HITI especially in terms of applicability to different mutations of the same gene (Discussion section, page 11).

12. How do your electroretinogram studies compared to what has been published for these mice?

Overall, our ERG results are similar to those published by Mao et al. (Hum Gene Ther. 2011 May;22(5):567-7) in the same animal model and at a similar timepoint (p90). A comment on this has been added to the Materials and Methods section, page 6.

Minor comments:

1. Per figure 1 endogenous promoter for the target gene is not the same as the promoter that drives exogenous Cas9 – redraw figure

We agree with the Reviewer that the previous figure was not clear, and we have modified it accordingly.

2. Provide adequate references in the introduction around mutation prevalence, e.g. 40% of RP, is AD? The next line should refer specifically to Americans of European origin, not all patients in the United States.

We have modified the text and references according to the Reviewers' recommendations (Introduction section, page 3).

Also please cite relevant references for your animal models.

We have checked all the references through the text as suggested by the Reviewer.

3. Express enzyme activity as percent of Wild type

We have reported in the text the serum enzyme activity in treated animals as a percentage of wild-type mice (Results section, page 7). We kept absolute values of enzymatic activity in the graph of Figure 3B so that the reader can have both sets of information.

4. Discuss why GAG's are not improved by p30 despite having added the enzyme on day 1 of life. Shouldn't it be enough time?

We believe that, although the treatment was administered at p1-2, the time necessary for SpCas9 expression, cleavage of the target locus, integration of the donor DNA and expression of the therapeutic transgene can impact on short-term phenotypical correction. Although we have shown that HITI efficiency does not vary from p15 onward, we believe that short-term ARSB expression at p30 might have been insufficient to significantly reduce GAG levels. As GAG clearance occurs progressively over time, longer-term ARSB expression than at p30 is needed to obtain the significant reduction of urinary GAGs observed at later timepoints.

5. Provide albumin levels for normal mice

We have measured albumin levels in untreated wild type (47 ± 5.4 , $n=7$) as well as MPS VI mice (39 ± 4.2 , $n=3$). As these are similar (p value = 0.70), we pooled them together in Figure 3I (also see the corresponding legend to the figure).

Reviewer #2 (Remarks to the Author):

In this paper the authors use homology-independent targeted integration (HITI) to both disrupt a mutant gene whilst inserting a therapeutic gene into a targeted locus using appropriate guide RNA sequences. The analyses are both in vitro and in vivo and tested in two organ systems: retina and liver. This is a very nice approach and one that may have widespread applications in CRISPR biology.

The authors present a huge amount of information and despite rereading it in detail several times, I was unable to follow the detail of what they were doing or why. For instance, there was no explanation that I could see as to why Kozak-DsRed or IRES-DsRed were compared.

We have modified the Discussion section (page 10) to explain the differences between the two translation start sites and the rationale for using them.

There are far too many acronyms which makes reading the text very difficult.

We agree that the text contains several acronyms and we looked at each of them to see if exchanging at least some with the full-length name can make the text easier to read. They are mostly technical and therefore their full-length word would make the text more difficult rather than easy to read; however, some of them were spelled out throughout the manuscript.

The data are presented without unoperated controls and so one cannot assess any potential therapeutic effect (in other words, the scrambled gRNA may simply be more toxic than the targeted gRNA sequences).

As subretinal injections per se cause damage of both retinal structure and function (Pawlyk et al., Hum Gene Ther. 2010 Aug;21(8):993-1004; Pang et al., Vision Res. 2008 Feb;48(3):377-85; Qi et al., PLoS One. 2015 Aug 28;10(8):e0136523), unoperated eyes are not the proper controls. To show that this holds through in our animal model, we have produced ERG data relative to either PBS- or scramble-injected mice and unoperated controls which show higher electrical responses in the latter and similar response between PBS- and scramble-injected eyes, as expected (see figure below). Therefore, we kept scramble-injected controls in Fig. 2C as these are better suited as negative controls to assess the therapeutic effect of the gRNA.

The combination of two organ systems is unusual and I wonder if it might be better to break this paper into two separate papers - one for eye and one for liver?

We agree that the proposed manuscript has the ambition of addressing different major limitations of AAV-mediated gene therapy in two different target tissues. However, as our take-home message is that the same platform (HITI) allows these different limitations to be overcome, we believe that they should be kept together. Indeed, other publications have shown the applicability of the same therapeutic platform (including HITI) in different tissues (Sukuzi et al., Nature. 2016 Dec 1;540(7631):144-149; Yao et al., Cell Res. 2017 Jun; 27(6):801-814).

In both cases the analyses require more depth. More detail on the ERG analyses (e.g. a wave, unoperated control)

As suggested by the Reviewer, we have added ERG data relative to the A-wave of the experimental eyes, which confirms the significant improvement observed at the B-wave level (Results section, page 5-6 and Fig. 2C). We have also measured visual acuity by Optometry, which shows significant improvement in the gRNA-treated eyes compared to the contralateral control eyes (Results section, page 6 and Fig. 2D). These results further support the therapeutic potential of AAV-HITI for dominant retinitis pigmentosa.

and better retinal histology might help in vivo studies

We have included histological images of better quality (see Fig. 2E).

and the liver work would be better assessed with a more thorough analysis of liver histology and enzymes.

As suggested by the Reviewer, we have performed a more thorough characterization of AAV-HITI efficacy in MPS VI mice. These results, which include liver ARSB activity as well as GAGs storage in visceral organs, in heart and in cortical bone have been added to the Results section, page 7 (Fig. 3). We have also performed preliminary safety studies that include liver histopathology and measurement of liver transaminases. None of these showed significant differences between gRNA- and scramble-treated controls. However, formal toxicity studies will be required if this approach will, at some point, translate to larger animals and humans.

The results are far too focussed on technical detail that should be supplemental and so the bigger picture of the message is lost.

We acknowledge this issue and have modified the text to reduce the technical details which have been moved to Supplementary Information. We hope the modified text will be clearer and easier to follow.

In brief, I have enjoyed reading the paper and this is a nice application of HITI which appears to work to some extent at least in vivo. I do not however feel qualified to give an opinion because I was unable to follow the detail of what was going on.

Reviewers' Comments:

Reviewer #1:

Remarks to the Author:

Summary:

This is an ambitious study trying to quantify and characterize two different disease models. While I understand the rationale of trying to show that HITI can address autosomal dominant retinal diseases and early-onset liver diseases, the overall effect is a manuscript that is hard to understand even for an expert. The manuscript leaves the reader feeling that neither disease is fully characterized and that critical questions are glossed over. I like the approach, and I believe it has some advantages, but I still have significant concerns.

Major comments.

1. One major issue is that the manuscript is hard to follow, and it takes a lot of effort to understand precisely what was done, how it was tested, and how significant the results are. This is due to ineffective and discursive writing and challenges with data presentation in the figures. There is a lot of back and forth between the main and supplementary figures. There are many experiments and information to convey, making it even more crucial that the text be clear and effectively guide the reader through the author's logic. Despite good work, this lack of clarity can result in an unfair evaluation of the manuscript.

2. The authors report performing deep sequencing analysis of 10 predicted off-target sites. However, the quantification in the retina/liver is not valid because your on-target is already very low, so it is very unlikely that you would see something with a lower frequency. The quantification is best done in the cells where you have high on-target rates. If I'm reading your data correctly, OFF-1 in the retina experiments is $\sim 1/7$ of your on-target site. That is very high!! This might be a moot point since you are characterizing a mouse guide and not a human guide to be used for therapeutic purposes, but it is important to clarify and address.

3. The translocation analysis is not quantitative at all.

4. Based on 2 and 3, I disagree with the author's conclusion that this is a thorough assessment of safety

5. It is not clear to me how significant this improvement is in the retina of RhoP23H-/+ mice. The differences look minimal, but I'm not an expert on these types of assays. To make the conclusion more convincing, it would help to include the data for wild-type mice as you did for MPS VI. It would also help to place the magnitude of the changes in the parameters against other therapies that have already been validated. For example, $\sim 5-10\%$ improvement in ERG (that is more or less what the figure shows) is sufficient to improve vision because...

6. Given how vital the skeletal phenotype is in MPS VI mice, I'm still wondering why the authors did not examine/quantify the bone abnormalities typical of this disease.

7. I reiterate my concern about understanding the potential outcomes of your gene modification strategy on a single-cell basis. This does not have to be done in the tissues, and it can be done in cells in culture. Your strategy can generate many different genotypes: Rho knockout, haploinsufficient, cells where the normal allele is knockout effectively only expressing the mutant allele, biallelic knock-in, in addition to all the various indels you can generate at the target site. It is possible that a considerable fraction of the modified cells could end up with a harmful combination of genotypes. Perhaps I am missing a reason why this is not relevant.

Minor comments.

1. Figure 1 is still challenging to understand. I imagine the black boxes and the figures represent Target sites? Part of what makes it difficult is that the arrows do not correspond to the experimental workflow in that both AAV's are delivered simultaneously.

2. How does mRNA or protein expression compare from the intact vs. inserted gene?

3. Include frequencies in the gated in the flow cytometry plots

4. It would help if the schematic for the AAV's were included in the main figures.

Reviewer #2:

Remarks to the Author:

The authors have addressed most of my points. The inclusion of the ERG data plots, as well as the optomotor responses add important validation to the claim of improved retinal function. The benefit is small, as expected with short term follow-up, but the point here is that it provides evidence for the AAV.HITI approach. The authors should be congratulated for developing such a complicated CRISPR system. I have no further requests to make.

REVIEWER COMMENTS

Reviewer #1 (Remarks to the Author):

Summary:

This is an ambitious study trying to quantify and characterize two different disease models. While I understand the rationale of trying to show that HITI can address autosomal dominant retinal diseases and early-onset liver diseases, the overall effect is a manuscript that is hard to understand even for an expert. The manuscript leaves the reader feeling that neither disease is fully characterized and that critical questions are glossed over. I like the approach, and I believe it has some advantages, but I still have significant concerns.

Major comments.

1. One major issue is that the manuscript is hard to follow, and it takes a lot of effort to understand precisely what was done, how it was tested, and how significant the results are. This is due to ineffective and discursive writing and challenges with data presentation in the figures. There is a lot of back and forth between the main and supplementary figures. There are many experiments and information to convey, making it even more crucial that the text be clear and effectively guide the reader through the author's logic. Despite good work, this lack of clarity can result in an unfair evaluation of the manuscript.

Based on Reviewer 2 comments on the previous version of our manuscript, we expanded the main text to include the data most relevant to the take-home message of our paper, which is the therapeutic relevance of HITI in mouse models of inherited diseases, while HITI molecular characterization (which includes efficiency of the cutting guides, and off-targets of both Cas9 and HITI) was moved to the Supplementary Results and Discussion section. For this reason, to avoid redundancy, reference in the main text to the Supplementary Results and Discussion section is only briefly made. We believe that this makes the main text easier to follow without too many technical details. Indeed, Reviewer 2 found this new version of our manuscript greatly improved. However, based on these last comments from Reviewer 1, we had two native English speakers, one of whom is a scientist, further reviewing the manuscript. They suggested minor modifications which we have made, but they found the overall structure easy to follow and that the message was clearly conveyed. However, if after these revisions, Reviewer 1 still thinks we should go back to the original structure of the manuscript where all sections are represented in the main text while the Supplementary parts contain the experiments mostly designed to address technical issues, we can do this.

2. The authors report performing deep sequencing analysis of 10 predicted off-target sites. However, the quantification in the retina/liver is not valid because your on-target is already very low, so it is very unlikely that you would see something with a lower frequency.

The on-target frequencies we have observed are very similar to those reported with HITI in liver by Suzuki *et al.* [Nature 2016 Dec 1;540(7631):144-149] and in retina by Nishiguchi *et al.* [Nat Commun 2020 Jan 24;11(1):482].

Specifically, the INDEL frequencies at the on-target sites detected by NGS were reported to be 19 and 4.5% in liver and retina, respectively, which are very close to those that we have observed in our study. In addition, the *Nature* study showed off-target INDEL frequencies similar to those that we found for the mAlb gRNA, leading us to the same conclusion about minimal off-target effects.

In addition, the Reviewer appreciates in a comment below that our analysis has allowed to identify off-targets that are less frequent than the on-target (OFF-1). Some of them are as frequent as 1/57 of the on-target (OFF-4), thus supporting that the sensitivity of our NGS analysis goes well beyond the detection of the on-target.

The quantification is best done in the cells where you have high on-target rates.

We believe that the quantification of off-target sites is more predictive *in vivo* than in immortalized, polyploid cell lines that have different proliferation rates, chromatin state/accessibility, and activities of the various DNA repair mechanisms than live tissues.

If I'm reading your data correctly, OFF-1 in the retina experiments is ~1/7 of your on-target site. That is very high!! This might be a moot point since you are characterizing a mouse guide and not a human guide to be used for therapeutic purposes, but it is important to clarify and address.

We agree with the Reviewer that the off-targets with highest frequencies (OFF-5 in addition to OFF-1) deserve a comment which we have since added to the Supplementary Results and Discussion section, page 1, lines 36-39. However, since both off-targets fall within intronic regions and are derived from a non-clinically relevant mouse guide, the Reviewer will agree with us that this should not represent a major concern at this stage.

3. The translocation analysis is not quantitative at all.

The translocation analysis is quantitative in principle, since we have designed a single PCR reaction to co-amplify both the wild-type and the rearranged alleles whose relative abundance can therefore be compared. Since no abnormal PCR products were detected, this comparison was clearly not possible.

Importantly, we acknowledge that rearrangements occurring with a frequency lower than the amplified HITI allele (therefore 5% and 4% for the retina and liver, respectively) are potentially not detectable with this assay. This is particularly true for insertions which will produce larger PCR products that can be more challenging to amplify than the HITI allele, while amplification of smaller products corresponding to deletions should be favored.

Overall, we can't exclude that any rearrangement occurring at a very low frequency might be undetectable with this assay and we pointed this out in the Supplementary Results and Discussion section, page 1, lines 42-43.

4. Based on 2 and 3, I disagree with the author's conclusion that this is a thorough assessment of safety.

Please see our answers above for 2 and 3. Nonetheless, we have mitigated our conclusion on this part and eliminated the statement that this is an in-depth demonstration of HITI safety. Please see the Supplementary Results and Discussion section, page 4, lines 172-174.

5. *It is not clear to me how significant this improvement is in the retina of $Rho^{P23H-/+}$ mice. The differences look minimal, but I'm not an expert on these types of assays. To make the conclusion more convincing, it would help to include the data for wild-type mice as you did for MPS VI.*

For all retinal functional and morphological analyses, data from age-matched wild-type mice are provided in the text of the manuscript. Please see the Results section, page 6, lines 179-182.

It would also help to place the magnitude of the changes in the parameters against other therapies that have already been validated. For example, ~5-10% improvement in ERG (that is more or less what the figure shows) is sufficient to improve vision because...

We observe an overall 18% ERG improvement given that the max b-wave amplitude response of the gRNA- and scramble-treated eyes are 349 μ V and 296 μ V, respectively. This is a significant improvement considering that, upon subretinal injection, about 30% of the retina is transduced while ERG records the whole retinal activity, from both treated and untreated areas. We acknowledge in the Discussion section (page 10, lines 317-321) that AAV-HITI results in a partial, yet significant, improvement of the retinal phenotype. Similar partial ERG improvements have been reported following either gene addition or other gene-editing approaches (Wagner et al., *Hum Gene Ther.* 2021 Sep 20; Patrizi et al., *Am J Hum Genet.* 2021 Feb 4;108(2):295-308; JCI Insight. 2017 Dec 21;2(24):e96560). Importantly, our data demonstrate that the ERG improvement is mirrored by a significant increase in visual acuity, as measured by Optomotry (Fig. 2D). This is not surprising since retinal signals are magnified by the visual pathway.

6. *Given how vital the skeletal phenotype is in MPS VI mice, I'm still wondering why the authors did not examine/quantify the bone abnormalities typical of this disease.*

We have extensive experience with both small (rats and mice) and large (cats) animal models of MPS VI as we have used them to test the efficacy of gene therapy (PMIDs: 17955027; 20021231; 21119624; 22428010; 23194248; 24725025; 27658524; 28932756) up to an ongoing phase I/II clinical trial (Clinicaltrials.gov Identifier: NCT03173521). In our assessments we have always characterized the skeletal phenotype of these models: while in MPS VI cats and rats, gene delivery performed early in life results in significant improvements to both bone length and histopathology [Tessitore *et al.*, *Mol Ther.* 2008 Jan;16(1):30-7; Cotugno *et al.*, *Hum Gene Ther.* 2010 May;21(5):555-69; Cotugno *et al.*, *Mol Ther.* 2011 Mar;19(3):461-9; Ferla et al., *Hum Gene Ther.* 2013 Feb;24(2):163-9], in MPS VI mice this did not occur, for neither gene or enzyme replacement therapy administered as early as newborn [Ferla *et al.*, *Hum Gene Ther.* 2014 Jul;25(7):609-18, and Figure below].

Femur length in MPS VI mice treated with neonatal ERT and AAV8.TBG.hARSB

The femur length was measured in 12-month-old MPS VI mice receiving weekly enzyme replacement therapy (ERT) starting from post-natal day (p) 2 and in normal (NR) and affected (AF) controls. Specifically, mice received either weekly ERT from p2 to p360 with (AAV+ERT) or without (ERT) a single systemic administration at p30 of 2×10^{12} gc/kg AAV8.TBG.hARSB or weekly ERT from p2 to p30 when they received 2×10^{12} gc/kg AAV8.TBG.hARSB (AAV). Values are reported as percentage (%) of age- and sex-matched littermate normal controls. Results are represented as a single measurement for each mouse (dot) and as mean \pm SEM for each group of treatment (column). Statistical comparisons were made using one-way ANOVA (p value < 0.0001) followed by the Tukey post-hoc test. **** = p < 0,0001.

This suggests that the long bone phenotype is hard to correct in this model. In line with this, we measured the femur length in one gRNA-treated mouse which was confirmed to be similar to that of a scramble-treated animal and shorter than that of an age- and sex-matched normal (NR) mouse, as expected (gRNA, 78% of NR; scramble, 83% of NR); based on this we did not pursue further long bone measurements in the current study and the animals were sacrificed for bone histological analyses.

The only long bone feature that we found improved in MPS VI mice following HITI was the osteocyte vacuolization in the femur and tibia cortical bone which we have now quantified as requested by the Reviewer (Results section, page 7, lines 240-241 and Figure 3E). On the other hand, articular and growth plate chondrocytes remained heavily vacuolized and disorganized (see Figure below).

Articular cartilage and growth plate did not improve in gRNA-treated mice.

Histological representative images of articular cartilage and growth plate of gRNA- and scramble-treated MPS VI affected mice and of normal (NR) controls; the black scale bar in the upper right image equals $75 \mu\text{m}$.

This is in line with the poor vascularization and low mannose receptor levels of articular cartilage and growth plate of MPS VI mice which explains why ERT improved bone-remodeling but not chondrocyte abnormalities and long bone growth in this model [Hendrickx *et al.*, Hum Mol Genet. 2020 Mar 27;29(5):803-816].

We have now mentioned the articular chondrocytes results, as data not shown, in the Results section, page 7, lines 241-242.

7. I reiterate my concern about understanding the potential outcomes of your gene modification strategy on a single-cell basis. This does not have to be done in the tissues, and it can be done in cells in culture. Your strategy can generate many different genotypes: Rho knockout, haploinsufficient, cells where the normal allele is knockout effectively only expressing the mutant allele, biallelic knock-in, in addition to all the various indels you can generate at the target site. It is possible that a considerable fraction of the modified cells could end up with a harmful combination of genotypes. Perhaps I am missing a reason why this is not relevant.

We agree with the Reviewer that this is an extremely relevant issue that we started to address in our previous revision (Fig. S3D), and which we have now implemented in order to define the relative abundance of the various genotypes produced by AAV-HITI. However, this has not been achieved either *in vitro* or by single-cell sequencing. We don't believe that addressing these issues *in vitro* is relevant because of the limitations imposed by the cell lines (which we have outlined in one of our previous answers). Instead, here we can take advantage of analyzing DNA extracted from tissues transduced *in vivo* which is much more relevant when considering *in vivo* gene therapy. We did not use single-cell sequencing for the technical challenge of isolating a sufficient number of single photoreceptors from a specific (transduced) area of the mouse retina. Even more importantly, the current design of our AAV-HITI constructs would allow the exclusive isolation of photoreceptors in which HITI has occurred, thus excluding from the evaluation those with different genotype outcomes, such as photoreceptors presenting INDELs alone.

Therefore, to define the genotypes produced by AAV-HITI, we started from the bulk NGS analysis of the transduced mouse retinal area whose results were presented in Figure S3D where we reported that INDEL accounted for 9%, HITI for 5% and wild-type for 86% of mRho alleles. To these allele frequencies we have applied the Hardy Weinberg equation to calculate the expected genotypic outcomes and found that 74% of cells will contain the unedited P23H/wild-type genotype. Of the remaining 26% edited cells: 13,2% will contain either the INDEL/wild-type or HITI/wild-type or INDEL/HITI or HITI/HITI genotypes, all expected to be therapeutic; 4,3% will be compound heterozygous for P23H/HITI which are predicted to be similar to the original P23H/wild-type genotype of our mouse model (defined as "no effect"); 7,7% will be P23H/INDEL which, based on the dominant negative effect of the P23H mutation reported by Rajan *et al.* [J Biol Chem 280, 1284–1291 (2005)], is expected to behave similarly to the original P23H/wild-type genotype (defined as "presumably no effect"); 0,8% will be INDEL/INDEL and thus defined as "deleterious", based on the severity of the mRho^{-/-} knock-out mouse model retinal phenotype compared to the phenotype of Rho^{P23H/+} mouse model [Humphries *et al.*, Nat Genet 15, 216–219 (1997); Jaissle *et al.*, Invest Ophthalmol Vis Sci 42, 506–513 (2001); Sakami *et al.*, J Biol Chem 286, 10551–10567 (2011)]. Therefore, only a minority of the genotypic outcomes of HITI will be deleterious with a significantly larger fraction being therapeutic. This is importantly reflected by the rescue observed in Rho^{P23H/+} mice following AAV-HITI.

In conclusion, we strongly believe that the analysis we have performed is a valid alternative to single cell sequencing providing similar information in terms of the genotypes produced.

This has been added to the Supplementary Results and Discussion section, page 2, lines 52-64 and schematically summarized in the current Table S3 which is shown below for clarity.

Predicted genotype frequency following HITI at the m*Rho* locus.

Genotype	Expected outcome	Frequency	Total
P23H/wild-type	Unmodified		74,0%
INDEL/wild-type	Therapeutic	7,7%	13,2%
HITI/wild-type	Therapeutic	4,3%	
INDEL/HITI	Therapeutic	0,9%	
HITI/HITI	Therapeutic	0,3%	
P23H/INDEL	Presumably no effect		7,7%
P23H/HITI	No effect		4,3%
INDEL/INDEL	Deleterious		0,8%

Minor comments.

1. Figure 1 is still challenging to understand. I imagine the black boxes and the figures represent Target sites? Part of what makes it difficult is that the arrows do not correspond to the experimental workflow in that both AAV's are delivered simultaneously.

We apologize for this, part of the figure legend was inadvertently left out. Accordingly, we have restored the legend to Figure 1 (page 4, line 125) and modified the figure to better reflect the experimental workflow. Please see the current Figure 1.

2. How does mRNA or protein expression compare from the intact vs. inserted gene?

This is an interesting point that in principle could be addressed by single-cell transcriptomics, keeping in mind the challenge of isolating single transduced photoreceptors. Additional challenges to this comparison are both the high nucleotide sequence homology (~90%) between the mouse and human orthologues, and the different affinity of antibodies directed to either mouse or human Rhodopsin. While we could not address experimentally this point, we should consider that HITI has been designed to occur near the promoter to guarantee high transgene expression levels, and we can infer that *Rhodopsin* is expressed at therapeutic levels from the significant improvement of the retinal phenotype that we observe.

3. Include frequencies in the gated in the flow cytometry plots

The percentage of EGFP+/DsRed+ cells is now included in the corresponding gate. Please see the current Figure S1.

4. It would help if the schematic for the AAV's were included in the main figures.

We have included the schematic of the AAV constructs in the main figures. Please see the resulting figures below which in our opinion are too crowded. Therefore, we suggest keeping the AAV constructs in the original Fig. S2 and S6. However, if the Reviewer and the Editor feel differently, we are willing to include the AAV constructs in the main figures as suggested.

Figure 2.

Figure 3.

Reviewer #2 (Remarks to the Author):

The authors have addressed most of my points. The inclusion of the ERG data plots, as well as the optomotor responses add important validation to the claim of improved retinal function. The benefit is small, as expected with short term follow-up, but the point here is that it provides evidence for the AAV.HITI approach. The authors should be congratulated for developing such a complicated CRISPR system. I have no further requests to make.

We thank the Reviewer for his/her positive comments.

Reviewers' Comments:

Reviewer #1:

Remarks to the Author:

The authors have improved the manuscript's readability and have included additional information to facilitate the interpretation of their findings. Some of the concerns on safety remain to be thoroughly characterized. This should not prevent publication if it is explicitly stated in the manuscript.

REVIEWERS' COMMENTS

Reviewer #1 (Remarks to the Author):

The authors have improved the manuscript's readability and have included additional information to facilitate the interpretation of their findings. Some of the concerns on safety remain to be thoroughly characterized. This should not prevent publication if it is explicitly stated in the manuscript.

We thank Reviewer 1 for his/her comments which helped to improve our manuscript. We have included in the Discussion section a statement that a thorough characterization of HIT1 is required before further translation.